# Explainable Token-level Noise Filtering for LLM Fine-tuning Datasets

**Yuchen Yang**[*,1,2], **Wenze Lin**[*,3], **Enhao Huang**[1,2], **Zhixuan Chu**[✉,1,2], **Hongbin Zhou**[4], **Lan Tao**[4], **Yiming Li**[5], **Zhan Qin**[1,2], **Kui Ren**[1,2]

[1]The State Key Laboratory of Blockchain and Data Security, Zhejiang University
[2]Hangzhou HighTech Zone (Binjiang) Blockchain and Data Security Research Institute, China
[3]Tsinghua University  [4]Alibaba Group  [5]Nanyang Technological University

[*] Equal contribution  [✉] Corresponding to: `zhixuanchu@zju.edu.cn`

## Abstract

Large Language Models (LLMs) have seen remarkable advancements, achieving state-of-the-art results in diverse applications. Fine-tuning, an important step for adapting LLMs to specific downstream tasks, typically involves further training on corresponding datasets. However, a fundamental discrepancy exists between current fine-tuning datasets and the token-level optimization mechanism of LLMs: most datasets are designed at the sentence-level, which introduces token-level noise, causing negative influence to final performance. In this paper, we propose `XTF`, an *explainable token-level noise filtering* framework. `XTF` decomposes the complex and subtle contributions of token-level data to the fine-tuning process into three distinct and explicit attributes (*reasoning importance*, *knowledge novelty*, and *task relevance*), which can be assessed using scoring methods, and then masks the gradients of selected noisy tokens accordingly to optimize the performance of fine-tuned LLMs. We conduct extensive experiments on three representative downstream tasks (math, code and medicine) across 7 mainstream LLMs. The results demonstrate that `XTF` can significantly improve downstream performance by up to 13.7% compared to regular fine-tuning. Our work highlights the importance of token-level dataset optimization, and demonstrates the potential of strategies based on attribute decomposition for explaining complex training mechanisms.

## 1 Introduction

LLM technology (An et al., 2024; Nam et al., 2024; Kambhampati et al., 2024) has developed rapidly in recent years, possessing powerful reasoning capabilities and enabling widespread application across various downstream tasks (Thirunavukarasu et al., 2023b; Wen et al., 2024; Guo et al., 2024). Meanwhile, to enhance the performance of LLMs in specific downstream tasks, developers usually train the base models, *i.e.*, general purpose LLM such as Llama (Touvron et al., 2023) and Deepseek (Guo et al., 2025), using relevant datasets, making adjustments to the model parameters. This technique, known as fine-tuning, is widely used in practical applications (Wang et al., 2022; Lin et al., 2024a).

However, existing fine-tuning datasets do not align fully with the token-by-token optimization process of LLMs. While LLM fine-tuning involves computing a loss at each token and updating model parameters accordingly, most fine-tuning datasets are designed at the sentence level, providing label sentences as the target output. Since not all tokens (in the label sentence) are valuable for performance improvement (Lin et al., 2024b; Peng et al., 2023), training in the entire label sentence can possibly introduce token-level noise and misguide the direction of convergence, ultimately reducing performance of fine-tuned LLMs in the target downstream task.

Current research lacks the capability to optimize datasets at the token level for LLM fine-tuning tasks. Mainstream data optimization methods fall into two categories: data filtering (Li et al., 2019; Goyal et al., 2024) and data augmentation (Dai et al., 2025; Ding et al., 2024). All of these approaches operate at the sample level and thus fail to further eliminate token-level noise. Some existing studies

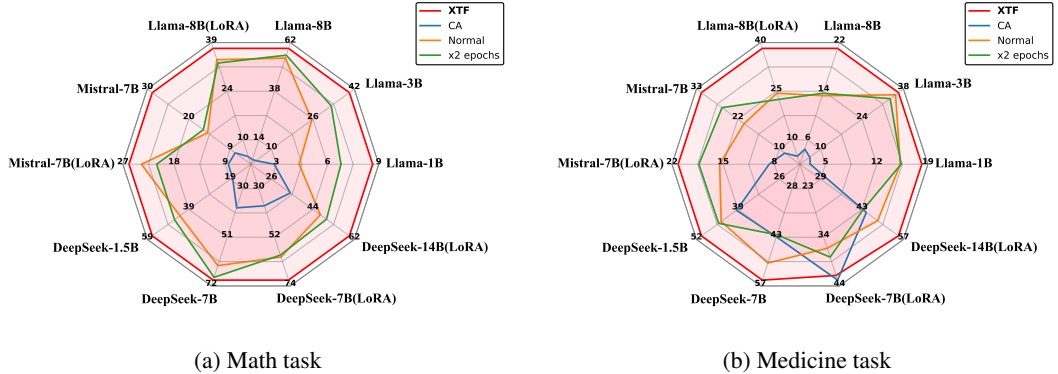

(a) Math task                    (b) Medicine task

Figure 1: The accuracy performance of our method on different LLMs. The results show that our method can significantly improve the final performance of fine-tuned LLMs across almost every case.

have explored the differences between token-level and sentence-level data from various perspectives, such as pretraining (Lin et al., 2024b), human preference optimization (Zeng et al., 2024; Yoon et al., 2024), and knowledge distillation (Wei et al., 2024; Cui et al., 2025). However, these works are often limited to specific scenarios (*e.g.*, pretraining with lower data quality requirements or direct preference optimization (DPO) (Rafailov et al., 2023) that relies on prior knowledge of labeled text pairs) or do not sufficiently investigate the value differences between tokens (*e.g.*, as in some knowledge distillation approaches (Peng et al., 2023; Cui et al., 2025)), rendering them unsuitable for fine-tuning dataset optimization.

Achieving token-level dataset optimization for fine-tuning necessitates filtering out tokens in output labels that do not contribute to final performance, which is a non-trivial task. Firstly, no existing research clearly elucidates the relationship between individual tokens within these labels and fine-tuning effectiveness. Although some explainability studies can identify connections between tokens in the input content and the correct generation of label sentences during the reasoning process (Chu et al., 2024; Zhao et al., 2024), they cannot explain the value of specific tokens in label sentences for fine-tuning tasks. Secondly, fine-tuning performance depends on both the base model's pre-existing knowledge and the specifics of the target task. When filtering noisy tokens, it is essential to consider both the base model's understanding of the data and that data's relevance to the downstream task (Liu et al., 2022b; Han et al., 2024). Therefore, filtering noisy tokens from fine-tuning datasets requires a comprehensive consideration of fine-tuning task requirements, rather than reliance on a single assessment criterion.

Motivated by the preceding discussion, we propose an **e**xplainable **t**oken-level data **f**iltering method XTF. This method aims to assess the value of token-level data within fine-tuning datasets and filter out noisy tokens by considering the specific characteristics of LLM fine-tuning. XTF consists of three phases. In the first phase, we decompose the contribution of data to the fine-tuning effect into three attributes (reasoning importance, knowledge novelty, and task relevance) to reduce the complexity of token value analysis. Concurrently, we define the criteria for identifying noisy tokens based on these attributes and provide theoretical justification in Appendix A. In the second phase, while considering the two key factors essential to fine-tuning: base model and task dataset, we design scoring mechanisms for the three attributes with controllable computational costs: **1) Reasoning Importance:** We concatenate the input and label output, then compute attention scores for each token using the base model. A lower attention score indicates a lower reasoning importance. **2) Knowledge Novelty:** We introduce the **p**robability of **c**orrect token **p**rediction (PCP) to quantify the novelty of knowledge learned from the fine-tuning dataset. A higher PCP indicates lower knowledge novelty. **3) Task Relevance:** We assess task relevance using embedding vectors generated by the base model for context-free inputs. The task domain is approximated by the average embedding of data samples, and token relevance scores are determined by their distance from the domain center. A larger distance implies lower task relevance. We present detailed scoring procedures in Section 3.2. In the third phase, we identify noisy tokens based on the statistical results and mask the gradients associated with these tokens during training to enhance the performance of the fine-tuned LLM. We adopt a conservative strategy to ensure the filtered tokens align with the criteria established in the first phase.

We conduct extensive experiments on datasets across 3 different downstream tasks and 7 representative LLMs. As shown in Figure 1, `XTF` can achieve up to 13.3% and 13.7% accuracy optimization on math task and medicine task, respectively, demonstrating its effectiveness on noise filtering and fine-tuning enhancement. In code generation, our method achieves performance optimization of up to 5.6%, 5.6%, and 6.3% on pass@1, pass@5, and pass@10, respectively, demonstrating that `XTF` is also effective in complex tasks involving multi-chance generation.

Our main contributions are three-fold:

- We reveal the research gap of token-level data optimization for LLM fine-tuning.
- We explore solutions for filtering token-level noise in fine-tuning datasets via three decomposed attributes: *reasoning importance*, *knowledge novelty*, and *task relevance*, proposing `XTF`.
- We conduct extensive experiments of `XTF` across multiple representative LLMs and downstream tasks, verifying its superior performance on fine-tuning optimization, demonstrates the potential of strategies based on attribute decomposition for explaining complex training mechanisms.

## 2 BACKGROUND AND RELATED WORK

### 2.1 LARGE LANGUAGE MODELS FINE-TUNING

Fine-tuning large models is widely recognized as a key technique to enhance their performance on downstream tasks (Wang et al., 2022; Han et al., 2024; Lin et al., 2024a). Numerous task-oriented models have been developed by fine-tuning high-performance base models, such as Llama-Math (Muñoz et al., 2024) and Llama-Finance (Cheng et al., 2024). The quality of the fine-tuning dataset is a critical factor that determines the effectiveness of this process (Zhou et al., 2023; Kuramoto & Suzuki, 2025). Therefore, optimizing data is essential for enhancing fine-tuning outcomes.

### 2.2 EXPLAINABILITY FOR LLMS

Explainability methods for LLMs aim to explain the decision-making mechanisms and behavioral logic of these models (Zhao et al., 2024; Chu et al., 2024). However, current techniques predominantly focus on the reasoning process (Huang et al., 2023; Kang et al., 2024; Singh et al., 2024), while research of explaining the relationship between tokens and fine-tuning outcomes remains limited.

### 2.3 TOKEN-LEVEL TRAINING

Several pioneering works have explored token-level data refinement in training, broadly categorized into three domains: data distillation, direct model training, and human preference training. **Data distillation** approaches (Wei et al., 2024; Cui et al., 2025; Liu et al., 2022a) primarily focus on the performance differences observed when a student model learns from token-level outputs (i.e., logits) versus sentence-level outputs, rather than specifically differentiating the value among individual tokens. **Direct model training**, including studies on training small-scale transformer models (Peng et al., 2023) and LLM pretraining (Lin et al., 2024b), utilizes changes in loss values to guide the selection of valuable tokens. We observe that these methods operate under an implicit assumption: that high-quality datasets are entirely noise-free and thus capable of correctly guiding token selection. This assumption rarely holds true in fine-tuning tasks, as base models often already exhibit strong performance, making it challenging to optimize dataset quality sufficiently to satisfy this noise-free criterion. Furthermore, existing works (Peng et al., 2023; Lin et al., 2024b) do not demonstrate that high-quality datasets are, in fact, approximately devoid of noisy tokens, which is a critical premise for these methods. **Human preference training** often involves specific optimization frameworks that rely on prior knowledge of labeled text pairs (Zeng et al., 2024; Xia et al., 2024; Yoon et al., 2024) and the construction of specialized token-level reward models; however, this approach can be challenging to generalize to broader application scenarios.

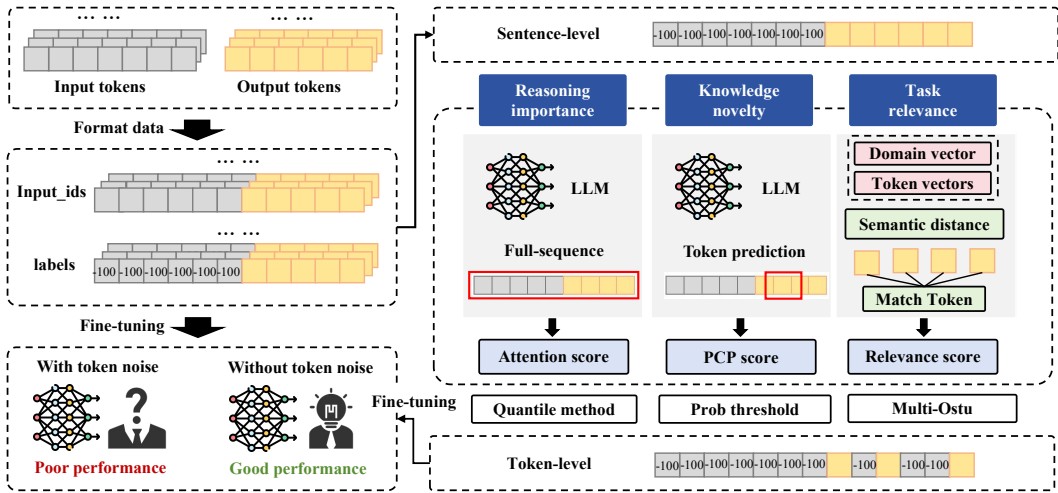

Figure 2: The pipeline of token-level data filtering comprises three steps. In the first step, we preprocess the dataset based on a regular format function. In the second phase, we get the sentence-level data item and assess three scores, *i.e.*, attention score, PCP score and relevance score, for the tokens of output label, suggesting the selection of noisy tokens. In the third phase, we mask the noisy tokens and fine-tune the target LLM.

## 3 METHODOLOGY

### 3.1 WHICH DATA ACTS AS NOISE FOR FINE-TUNING?

Due to scale differences, it is inherently difficult to intuitively assess the impact of token-level data on the final fine-tuning outcome. Therefore, we attempt to explain the contribution of token data to fine-tuning from three attributions through theoretical analysis.

A fine-tuning process can be conceptualized as an alignment between a high-performance base model and a task-specific dataset. Consequently, the performance of the fine-tuned model should be influenced by three factors: the cognition of the base model, the knowledge in the task dataset, and the contradiction between the base model and the task dataset. When we aim to mask a token from the label sentence, we can assess its potential impact on the fine-tuning result from these three perspectives. Specifically, we propose three attributes that positively influence the fine-tuning process: for the cognition of the base model, we extract **reasoning importance**; for the discrepancy between the base model and the task dataset, we extract **knowledge novelty**; and for the knowledge in the task dataset, we extract **task relevance**. These attributes represent the following meanings:

**Reasoning Importance (RI)**: whether the presence or absence of this token significantly affects the base model's inference results;

**Knowledge Novelty (KN)**: whether the presence of this token is novel to the base model;

**Task Relevance (TR)**: whether the presence of this token is related to the objective of the task dataset.

It is not feasible to consider all three properties simultaneously and assign a composite score, as there is no clear basis to determine their interrelationship or any hierarchical order. However, we can still use these three properties to identify which tokens are noise. Specifically, we find that if a token completely lacks any of the three attributes, it can be considered as noise. This is intuitive; for example, if a token is entirely unrelated to the task objective (lacking the TR attribute), then even if it may influence the base model's subsequent generation results (RI) or represent a prediction that the base model has not yet learned (KN), it does not contribute to the fine-tuning task. We elaborate on the analysis of the three derived attributes and formally prove the correctness of this judgment through logical reasoning in the Appendix A.

Formally, the token-level noise in the fine-tuning tasks can be represented as:

$$D_{\text{noise}} = (D_{RI\downarrow}) \cup (D_{KN\downarrow}) \cup (D_{TR\downarrow}) \tag{1}$$

where $(D_{RI\downarrow})$, $(D_{KN\downarrow})$ and $(D_{TR\downarrow})$ respectively represent data lacking reasoning importance, knowledge novelty, and task relevance.

## 3.2 How to assess Tokens?

After obtaining the three attributes and defining noise data, we employ a parameterized scoring mechanism to separately assess the three attributes and to identify noise tokens in the dataset. In alignment with real-world requirements, the choice of scoring method is expected to meet two requirements: **1) Controllable computational cost:** Since the dataset contains a vast number of tokens, the assessment method should not impose excessive computational overhead. **2) Joint consideration of the base model and the data**: The assessment of the three attributes must not be separated from the essential elements of the fine-tuning task itself. Consequently, we adopt three reasoning-level explainability methods to assess the three attributes separately.

**Attention Score for RI.** We employ the base model's attention scores to assess reasoning importance. Attention is a mechanism that adaptively learns the contextual importance of tokens during the pretraining of the base model (Vaswani et al., 2017). Existing work has demonstrated that masking low-attention tokens during the LLM reasoning process can even enhance generation quality (Gupta et al., 2021). Therefore, using attention scores to assess reasoning importance aligns with our needs. We input the entire text (including both the input sentence and the output label sentence) into the model and compute the attention scores. We formulate the reasoning importance score $\mathcal{S}_{\mathrm{RI}}$ for the $k$-th token in the output label sentence as:

$$\mathcal{S}_{\mathrm{RI}}(O_k) = \mathcal{A}\left(\theta, I + O\right)\left[l_I + k\right],\tag{2}$$

where $\theta$ denotes the parameters of base model, $\mathcal{A}$ is the function to compute attention value, $I$, $O$ represents the input tokens and output label respectively, and $l_I$ is the length of input tokens. This approach can be applied to explain the tokens in the output label and considers the base model's understanding of the data throughout the answer generation logic.

**PCP Score for KN.** For knowledge novelty, we adopt the base model's **p**robability of **c**orrect token **p**rediction (PCP). Intuitively, a lower probability indicates reduced confidence in predicting the token, suggesting that it is more likely to contain knowledge that the model has not yet acquired. We formulate the knowledge novelty score $\mathcal{S}_{\mathrm{KN}}$ as:

$$\mathcal{S}_{\mathrm{KN}}(O_k) = 1 - P\left(O_k \mid I + [O_0, O_1, \ldots, O_{k-1}]\right).\tag{3}$$

**Distance Score for TR.** For relevance scoring, we calculate semantic distance based on the base model's embedding layer. First, we feed each data item from the dataset into the base model in its entirety and obtain the average value of the embedding layer vectors as the domain vector for the fine-tuning task. Then, we collect all tokens appearing in the dataset and compute their context-free embedding vectors. Finally, we measure the distance between these token vectors and the domain vector, using this distance as the relevance score. The specific formulas for the task relevance scores are as follows:

$$\mathcal{V}(\text{Domain}) = \frac{\sum\left(\mathcal{E}(\theta, exp_w)\right)}{n_w},$$
$$\mathcal{S}_{\mathrm{TR}}(O_k) = 1 - \text{Normalize}\left(\mathcal{D}\left(\mathcal{E}(O_k), \mathcal{V}(\text{Domain})\right)\right),\tag{4}$$

where $\mathcal{E}$ denotes the function to get the embedding vector of inputs, $exp_w$ and $n_w$ represents the expert words and its number, respectively. $\mathcal{D}$ is the function to compute the distance of two vector.

## 3.3 How to Fine-tune based on Scores?

In this section, we will describe how to correctly filter noisy tokens using a conservative strategy and ignore them during the fine-tuning process through gradient masking.

**Token Filtering.** After obtaining scores for each dimension, we need to filter the noisy tokens based on the scores. As shown in Figure 3, the distributions of scores across different tasks share the same features, requiring us to design adaptive mechanism for each attributes. Specifically, the

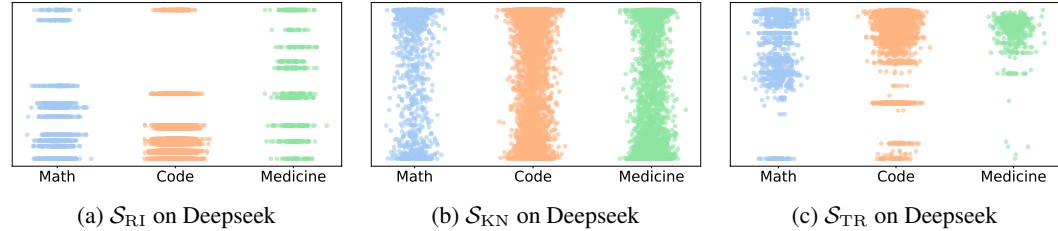

(a) $\mathcal{S}_{\mathrm{RI}}$ on Deepseek  (b) $\mathcal{S}_{\mathrm{KN}}$ on Deepseek  (c) $\mathcal{S}_{\mathrm{TR}}$ on Deepseek

Figure 3: Distribution of the three scores across different datasets on Deepseek-1.5B. The reasoning importance score is distributed across some fixed values, the knowledge novelty score has a somewhat uniform distribution, and the task relevance score's distribution exhibits clustering features. More distribution figures of different LLMs are shown Appendix D.1.

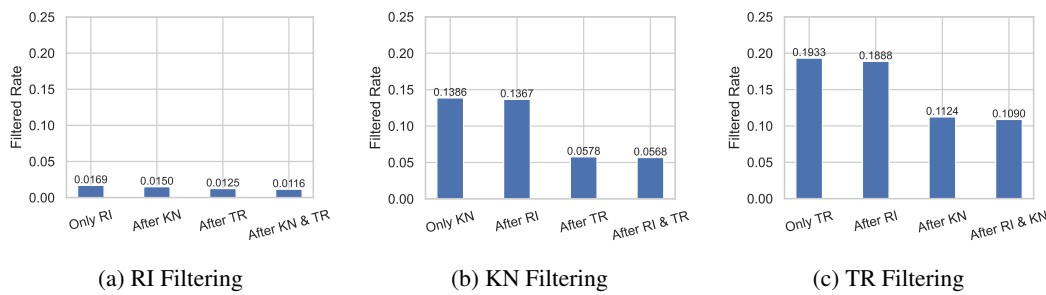

(a) RI Filtering  (b) KN Filtering  (c) TR Filtering

Figure 4: Complementarity among attributes using Deepseek-1.5B and GSM8k. Using RI as an example, Only RI represents the percentage of tokens that can be filtered using only RI. After KN represents the percentage of tokens that RI can filter after KN is applied. The reduction in tokens when comparing After KN with Only RI indicates the tokens on which KN and RI overlap.

reasoning importance scores exhibit an extreme distribution where many scores share identical values and demonstrate significant differences after normalization. We directly apply the quantile method (Interquartile Range) to filter out tokens with extremely low scores. In mathematical expression:

$$
\begin{aligned}
Q_1, Q3 &= \mathrm{Quantile}(S_{RI}(O), [25, 75]), \\
IQR &= Q_1 - (Q_3 - Q_1), \\
O_k &\in (D_{RI\downarrow}) \text{ if } S_{RI}(O_k) < Q_1 - IQR,
\end{aligned}
\tag{5}
$$

where $S_{RI}(O)$ means all the reasoning importance score of tokens in output label $O$. The knowledge novelty scores display a uniform distribution, which makes it difficult to distinguish low scores. Therefore, we adopt a heuristic threshold. We consider tokens with a PCP higher than 95% as only containing knowledge without novelty and treat them as noise. In mathematical expression:

$$
O_k \in (D_{KN\downarrow}) \text{ if } S_{KN}(O_k) < 0.05.
\tag{6}
$$

Finally, the task relevance scores exhibit cluster characteristics, for which we employ the Multi-Otsu method (Liu & Yu, 2009) to partition the scores. Since the cluster with the smallest mean value typically consists of space replacement symbols, we filter out the tokens in the cluster with the second smallest mean value. In mathematical expression:

$$
O_k \in (D_{TR\downarrow}) \text{ if } S_{TR}(O_k) \in \mathcal{M}(S_{TR})^{2nd},
\tag{7}
$$

where $\mathcal{M}$ is the Multi-Otsu method and $\mathcal{M}(S_{TR})^{2nd}$ denotes the cluster with the second smallest mean value. The detail of Multi-Otsu method is shown in Appendix B.

**Threshold Analysis.** XTF adopts an aggressive filtering strategy, namely, it takes the union of three types of filtered tokens. The premise that supports this strategy is that the filtered tokens are assumed to be "completely" devoid of the corresponding attributes. Therefore, tokens with significantly low scores should be selected for filtering, which is also why a conservative threshold is chosen. Meanwhile, within the XTF framework, filtering attributes across multiple dimensions provide complementary effects, compensating for noise omissions introduced by the loose threshold.

Figure 4 illustrates the complementarity among attributes: the overlap of tokens filtered by different attributes is not higher than 58.3%. This strategy is based on a simple yet useful heuristic: ambiguous noise that is difficult to distinguish from one perspective can be clearly identified from another perspective. When noise is jointly evaluated from multiple perspectives, a complex problem is transformed into several simpler problems. To further explore the results of token filtering, we have provided specific examples of token filtering in Appendix F.

**Training the Model.** Here we describe how to optimize fine-tuning based on token filtering results. As shown in Figure 2, when processing the data, because the fine-tuning task focuses solely on the correctness of the output, we mask the input tokens by assigning them a default value (often *[-100]*) and thereby exclude them from gradient computation. After identifying noisy tokens in the output labels, we mark them with the this default value and use the resulting data for fine-tuning. Given the noisy token list $N$, the loss function $\mathcal{L}_F$ for learning a data item can be expressed as:

$$\mathcal{L}_F = - \sum_{O_k \notin N} \log P\big(O_k \mid I + [O_0, O_1, \ldots, O_{k-1}]\big). \tag{8}$$

## 4 EXPERIMENTS

### 4.1 EXPERIMENT SETTINGS

**Dataset.** We select three representative downstream tasks to evaluate the fine-tuning performance, including two mainstream tasks: math, and code (widely used to evaluate LLMs (Guo et al., 2025; OpenAI et al., 2024; Team et al., 2024)), as well as an important specialized tasks: medicine (Thirunavukarasu et al., 2023a; Alberts et al., 2023). For the math task, we employ the GSM8K (Cobbe et al., 2021) for fine-tuning and evaluation. For the code task, we fine-tune on the Code-Exercise (AI, 2025) and evaluate using HumanEval (Chen et al., 2021). For medicine tasks, we employ the PubMedQA (Jin et al., 2019) for fine-tuning and evaluation. We also conduct additional experiments based on the NuminaMath-CoT (Li et al., 2024), MATH-500 (Hendrycks et al., 2021), and FIQA (Maia et al., 2018) datasets, but due to space limitations, these results are presented in the Appendix C.1.

**LLMs.** We select 7 different base LLMs of varying scales from three outstanding model families: DeepSeek (Guo et al., 2025), Llama (Touvron et al., 2023; Grattafiori et al., 2024) and Mistral (Jiang et al., 2023). Specifically, for the Deepseek family, we choose DeepSeek-R1-distilled-Qwen-1.5B, 7B and 14B (Bai et al., 2023). For the Llama family, we select Llama-3.2-1B, 3B and Llama-3.1-8B. For the Mistral family, we select Mistral-v0.1-7B.

**Baselines.** We consider 3 regular LLM implementations and 4 data enhancement methods to demonstrate the effectiveness of our `XTF`. Specifically, for regular LLM implementations, we adopt clean accuracy (CA), i.e., using the original base model, normal fine-tuning (Normal) and more epochs ($\times 2$ epochs). Data enhancement methods can filter the data noise from different perspectives. Specifically, data filtering (DF) (Li et al., 2019) filters noisy data at the sample level, data augmentation (DA) (Dai et al., 2025) augments more data to enhance the model's robustness, thus resist noise, selective language model training (Lin et al., 2024b) trains the token-level data selectively based on changes in the loss value, and token cleaning (TC) (Pang et al., 2025) performs fine-grained data selection for LLM supervised fine-tuning. More details are shown in Appendix B.2.

**Metrics.** We primarily use accuracy to evaluate the performance of LLMs on specific tasks. For tasks with a standard answer, we adopt a zero-shot form to pose the questions and use a judge model (Anthropic, 2025) to determine the correctness of the responses. This evaluation method is less influenced by the format of the prompt and the model's response style, providing a clear and intuitive reflection of the fine-tuning effect. For the code completion task, we assess performance using pass@1, pass@5, and pass@10 metrics, which are specifically used to evaluate code generation tasks (Kulal et al., 2019).

**Hyperparameter.** We conduct experiments based on existing work and strictly control the fairness of the results through hyperparameters, as detailed in Appendix B.3.

Table 1: Result of main experiment. We show the accuracy of LLMs across different fine-tuning methods. Best results are marked in **bold** and the second best results are marked with underline.

| MATH: Fine-tuning and evaluate models on GSM8K | | | | | | | | | | |
|---|---|---|---|---|---|---|---|---|---|---|
| **Model** | $|\theta|$ | **LoRA** | **CA** | **Normal** | **×2 Ep** | **DF** | **DA** | **SLM** | **TC** | **XTF** |
| Llama-3.2 | 1B | × | 2.8 | 4.3 | 6.8 | 7.6 | 2.4 | 5.9 | 6.8 | **8.7** |
| Llama-3.2 | 3B | × | 3.9 | 25.8 | 33.4 | 36.9 | 27.1 | 38.8 | 38.4 | **40.5** |
| Llama-3.1 | 8B | × | 4.6 | 54.0 | 55.4 | 52.7 | 55.4 | **60.3** | 58.9 | 58.7 |
| Llama-3.1 | 8B | ✓ | 4.6 | 33.7 | 32.7 | 37.0 | 33.7 | 37.9 | **38.4** | 37.1 |
| Mistral | 7B | × | 8.0 | 15.0 | 16.1 | 21.3 | 18.4 | 22.6 | 24.1 | **29.1** |
| Mistral | 7B | ✓ | 8.0 | 23.4 | 20.7 | 25.6 | 21.8 | 21.3 | 20.7 | **25.6** |
| Deepseek-distilled-qwen | 1.5B | × | 17.6 | 42.9 | 45.5 | 47.0 | 37.4 | 37.3 | 38.5 | **56.2** |
| Deepseek-distilled-qwen | 7B | × | 37.9 | 63.0 | 68.1 | 65.5 | 56.5 | 63.8 | 61.9 | **69.3** |
| Deepseek-distilled-qwen | 7B | ✓ | 37.9 | 61.3 | 60.4 | 67.9 | 62.0 | 68.2 | 70.0 | **71.8** |
| Deepseek-distilled-qwen | 14B | ✓ | 34.5 | 47.6 | 50.3 | 52.4 | 50.5 | 49.3 | 52.1 | **60.3** |
| Average | – | – | 16.0 | 37.1 | 39.0 | 41.4 | 36.5 | 40.5 | 41.0 | **45.7** |
| **MEDICINE: Fine-tuning and evaluate models on PubMedQA** | | | | | | | | | | |
| **Model** | $|\theta|$ | **LoRA** | **CA** | **Normal** | **×2 Ep** | **DF** | **DA** | **SLM** | **TC** | **XTF** |
| Llama-3.2 | 1B | × | 2.9 | 15.5 | 15.6 | 15.4 | 18.3 | 13.1 | 12.4 | **18.5** |
| Llama-3.2 | 3B | × | 6.6 | 35.5 | 33.7 | 29.8 | **37.9** | 36.1 | 36.4 | 36.5 |
| Llama-3.1 | 8B | × | 4.9 | 13.6 | 14.0 | 18.4 | 14.7 | **22.9** | 22.9 | 21.3 |
| Llama-3.1 | 8B | ✓ | 4.9 | 24.2 | 22.3 | 34.3 | 31.4 | 26.3 | 27.1 | **37.9** |
| Mistral | 7B | × | 8.6 | 20.0 | 26.2 | 18.7 | 23.4 | 26.5 | 26.2 | **32.0** |
| Mistral | 7B | ✓ | 8.6 | 15.5 | 18.4 | 15.6 | 13.1 | 17.5 | 16.9 | **21.3** |
| Deepseek-distilled-qwen | 1.5B | × | 39.8 | 44.6 | 45.4 | 41.2 | 49.7 | 48.3 | **51.2** | 50.8 |
| Deepseek-distilled-qwen | 7B | × | 42.7 | 50.5 | 42.1 | 55.4 | 52.7 | 51.3 | 51.6 | **55.6** |
| Deepseek-distilled-qwen | 7B | ✓ | **42.7** | 35.9 | 37.8 | 33.6 | 38.4 | 39.0 | 37.4 | 41.7 |
| Deepseek-distilled-qwen | 14B | ✓ | 44.6 | 48.5 | 43.4 | 51.3 | 47.0 | 53.9 | 54.6 | **55.7** |
| Average | – | – | 20.6 | 30.4 | 29.9 | 31.4 | 32.7 | 33.5 | 33.7 | **37.1** |

## 4.2 MAIN EXPERIMENT RESULTS

We employ a significant number of fine-tuning cases and three representative downstream tasks in our main experiment. By conducting fine-tuning experiments on different models using unified hyperparameters and datasets, and comparing the performance of the fine-tuned models, we can assess the effectiveness of the fine-tuning methods. During the training process, we employ a training set, validation set, and test set to prevent issues such as overfitting caused by varying convergence speeds. Specifically, the model parameters that perform best on the validation set are retained as the final model parameters and tested on the test set. The reported results are test set accuracy.

**Math.** Math is an important downstream task in LLM research. It is widely adopted as the LLM benchmark, and the performance on math task can reflect the logic ability of LLMs. As shown in Table 1, XTF has average 8.6% higher accuracy than normal fine-tuning and 4.3% higher accuracy than the best baseline DF. In all 10 cases, XTF obtains 8 best results and 2 second best results. In particular, when fine-tuning the Deepseek-distilled-qwen-1.5B with all parameters, XTF achieves 13.3% higher accuracy than normal fine-tuning and 9.3% higher accuracy than the best baseline DF. Due to space limitations, additional experiments on math tasks (NuminaMath-CoT and MATH-500) will be presented in the Appendix C.1.

**Medicine.** Medicine is a promising application area for LLMs, and LLM researches on pharmaceutical downstream tasks (Thirunavukarasu et al., 2023a; Alberts et al., 2023) have already achieved widespread influence. As shown in Table 1, XTF has average 6.7% higher accuracy than normal fine-tuning and 3.4% higher accuracy than the best baseline TC. In all the 10 cases, XTF obtains 6 best results and 4 second best results. When fine-tuning the Llama-3.1-8B with LoRA, XTF demonstrates 13.7% higher accuracy than normal fine-tuning and 3.6% higher accuracy than the best baseline.

**Code.** The code task is a more challenging task, and we provide pass@1, pass@5, and pass@10 results. Due to the generally lower accuracy on code tasks, the gap between XTF and the baseline is

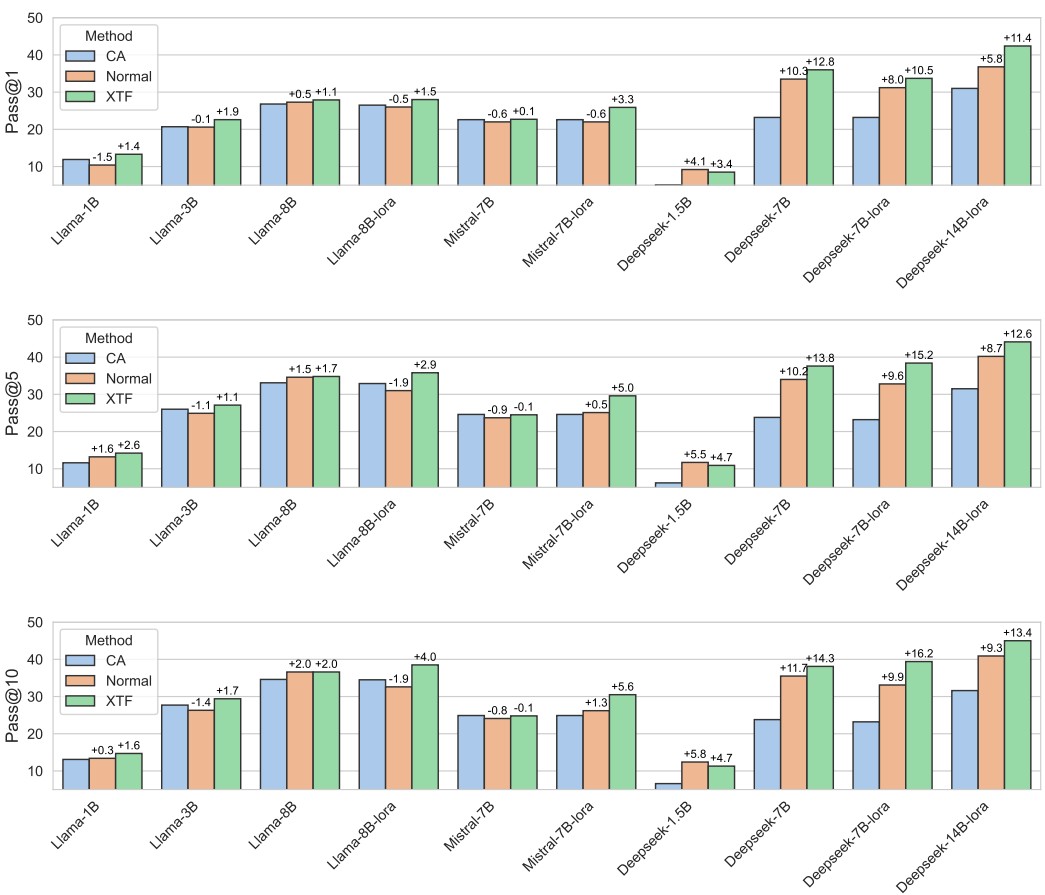

Figure 5: The results on code task. We show the results of pass@1, pass@5 and pass@10 respectively.

not as significant as in other experiments. As shown in Figure 5, `XTF` generally exhibits better results than normal fine-tuning. This difference increases when the LLMs are given more generation chances (from pass@1 to pass@10). In certain cases, normal fine-tuning decreases accuracy, indicating the harmfulness of noisy tokens, while `XTF` still shows positive performance. We find that when fine-tuning larger-scale models, the effect of noise filtering is more pronounced than when fine-tuning smaller-scale models. This phenomenon aligns with the claim we propose in Section 3.1: finding noise in data should consider the base model's knowledge. Since larger-scale models generally possess stronger performance, we believe that stronger base model performance can better leverage the `XTF` method.

## 4.3 ABLATION STUDY

In this section, we conduct an ablation study on three noise filtering attributes. We selectively use the attributes to filter the noisy tokens and train models from different series.

As shown in Table 2, `XTF` consistently demonstrates the best performance compared to other settings, suggesting that all the attributes are necessary for better noise filtering. At the same time, we observe that in mathematical tasks, using the combination of RI and KN to filter tokens is consistently superior to other combinations, whereas this is not the case in the Medicine task. Additionally, in the Medicine task, the optimal combination of attributes differs across models. These phenomena suggest that the relative effectiveness of the three attributes can vary depending on the model and task type, which aligns with our understanding of fine-tuning discussed in Section 3.1.

In addition, we also conduct ablation study against the threshold (for token filtering) selection, which are detailed in the Appendix C.2.

Table 2: Ablation study of `XTF` across different settings. DS, LA and MS denotes Deepseek, Llama and Mistral employed in this experiment. Ma and Me denotes math task and medicine task respectively. $\times$ means the corresponding noise has been filtered while $-$ means not. Best results are marked in **bold** and the second best results are marked with underline.

| Case | $D_{RI\downarrow}$ | $D_{KN\downarrow}$ | $D_{TR\downarrow}$ | DS(Ma) | LA(Ma) | MS(Ma) | DS(Me) | LA(Me) | MS(Me) | Avg |
|------|------|------|------|------|------|------|------|------|------|------|
| Zero | $-$ | $-$ | $-$ | 42.9 | 25.8 | 15.1 | 44.6 | 35.5 | 20.0 | 30.7 |
| I | $\times$ | $-$ | $-$ | 44.0 | 28.3 | 16.6 | 44.7 | 36.1 | 22.3 | 32.0 |
| II | $-$ | $\times$ | $-$ | 48.1 | 28.4 | 20.2 | 46.8 | 36.2 | 27.1 | 34.5 |
| III | $-$ | $-$ | $\times$ | 45.3 | 30.1 | 17.7 | 45.7 | 35.5 | 25.4 | 33.3 |
| IV | $\times$ | $\times$ | $-$ | 48.3 | 32.2 | 23.9 | 47.8 | 36.4 | 28.1 | 36.1 |
| V | $\times$ | $-$ | $\times$ | 49.2 | 34.1 | 27.7 | 47.1 | 35.9 | 27.6 | 36.9 |
| VI | $-$ | $\times$ | $\times$ | 47.3 | 32.9 | 22.6 | 48.3 | 36.1 | 30.9 | 36.3 |
| XTF (Ours) | $\times$ | $\times$ | $\times$ | **56.2** | **40.5** | **29.1** | **50.8** | **36.5** | **32.0** | **40.1** |

## 5 Discussion

Our proposed `XTF` effectively enhances LLM fine-tuning, but it still has some limitations. First, regarding computational cost, `XTF` incurs inference-level computational overhead (detailed in the Appendix D), which performs better compared to existing methods that train a reference model. However, it still imposes a significant burden when dealing with large models. If a small distilled model could be provided to identify noise for large-scale base models, it would significantly reduce the cost of token scoring. Additionally, we believe that more attributes can be explored for filtering noise, and these attributes can be assessed from multiple perspectives. Such work would provide an effective complement to the application of `XTF` in real-world scenarios.

## 6 Conclusion

In this paper, we investigated the influence of training data on fine-tuning performance at the token-level. We explored solutions for filtering token-level noise to optimize fine-tuning datasets using three decomposed dimensions: reasoning importance, knowledge novelty, and task relevance, subsequently proposing `XTF`. We conducted extensive experiments on datasets across 3 different downstream tasks and 7 representative LLMs. `XTF` achieved up to 13.3%, 13.7% and 6.3% accuracy optimization on the math task, medicine task and code task, respectively, and outperformed all the baselines overall, demonstrating its effectiveness in noise filtering and fine-tuning enhancement.

### Acknowledgements

This work was supported in part by the National Natural Science Foundation of China (Grant Nos. 62502435, 625B1032, 62441238, U2441240), the Zhejiang Provincial Natural Science Foundation (No. LQN26F020002), the "Pioneer" and "Leading Goose" R&D Program of Zhejiang (No. 2024C01169), and the Kunpeng-Ascend Science and Education Innovation Excellence/Incubation Center.

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

## A  THEORETICAL FOUNDATIONS OF XTF

**Executive summary.** We provide a complete, assumption–transparent, and *parameterization–invariant* analysis of how token filtering improves the alignment (in a Riemannian sense) between the training direction and the "ideal" gradient direction. We (i) clarify the geometry and regularity of the Fisher information (via damping), (ii) decouple the analysis from any specific optimizer by working with an *arbitrary* symmetric positive–definite (SPD) preconditioner $M$ (with $M = F_\lambda$ recovering natural gradient and $M = I$ recovering Euclidean SGD), (iii) make precise the statistical assumption on the *selector* (and quantify the effect of selection bias), (iv) replace global smoothness by *local* smoothness, and (v) tighten the treatment of high–confidence (KN) tokens with a clean and fully rigorous bound. Throughout, all random variables are defined on a common probability space; expectations are with respect to the distributions explicitly indicated.

## A.1 PRELIMINARIES AND GEOMETRY

**Contexts and model.**  A token–level *context* is $c := (x, t_{<i})$ and the associated label (gold token) is $t$. Let $p_\theta(t \mid c)$ denote the model's conditional distribution at parameters $\theta$, and let

$$\phi_\theta(c, t) := \nabla_\theta \log p_\theta(t \mid c)$$

be the *score* of the pair $(c, t)$.

**Teacher–forcing context law.**  Filtering will *remove* some token–level losses but, under teacher forcing, it does not alter the forward contexts $c$. We therefore fix a baseline context distribution

$$Q(c) \quad \text{(the teacher–forcing context law; e.g. the empirical distribution over contexts in the corpus).} \tag{9}$$

All expectations below that involve $p_\theta(\cdot \mid c)$ are taken with $c \sim Q$ first.

**Damped Fisher information.**  Define the (damped) conditional Fisher information at $\theta$:

$$F_\lambda(\theta) := \mathbb{E}_{c \sim Q}\, \mathbb{E}_{t \sim p_\theta(\cdot \mid c)}\big[\phi_\theta(c, t)\, \phi_\theta(c, t)^\top\big] + \lambda I, \qquad \lambda \geq 0. \tag{10}$$

We will assume $F_\lambda(\theta) \succ 0$ for all $\theta$ under consideration; see Assumption 1. Given any SPD matrix $M(\theta) \succ 0$ (to be specified below), we write

$$\langle u, v \rangle_M := u^\top M(\theta) v, \qquad \|u\|_{M^{-1}} := \sqrt{u^\top M(\theta)^{-1} u}.$$

The geometric viewpoint will be stated for a *general* preconditioner $M$; the *natural gradient* corresponds to the special choice $M = F_\lambda$.

**Ideal risk.**  Let $p_\star(c, t)$ denote the (unknown) *ideal* task distribution over $(c, t)$, with marginal $Q(c)$ under teacher forcing. The ideal per–token KL risk is

$$\mathcal{L}_\star(\theta) := \mathbb{E}_{(c,t) \sim p_\star}\big[-\log p_\theta(t \mid c)\big] = \mathrm{KL}\big(p_\star \,\|\, p_\theta\big) + \text{const.} \tag{11}$$

Since $p_\star$ does not depend on $\theta$,

$$\nabla_\theta \mathcal{L}_\star(\theta) = -\mathbb{E}_{(c,t) \sim p_\star}\big[\phi_\theta(c, t)\big]. \tag{12}$$

For any SPD $M$, we write the (preconditioned) gradient direction as $\widetilde{\nabla}_M \mathcal{L}_\star(\theta) := M(\theta)^{-1} \nabla_\theta \mathcal{L}_\star(\theta)$; for $M = F_\lambda$ this is the damped natural gradient.

## A.2 DATA MODEL, SELECTOR, AND ASSUMPTIONS

**Assumption 1** (Damped Fisher and teacher forcing). (1) $Q(c)$ is fixed by teacher forcing and independent of filtering. (2) There exists $\lambda \geq 0$ such that $F_\lambda(\theta) \succ 0$ for all $\theta$ in a neighborhood of interest (e.g. along the optimization trajectory).

**Assumption 2** (Mixture model of labels). There exists $\varepsilon \in [0, 1)$ and distributions $p_{\text{core}}, p_{\text{noise}}$ on $(c, t)$ such that

$$p_{\text{train}}(c, t) = (1 - \varepsilon)\, p_{\text{core}}(c, t) + \varepsilon\, p_{\text{noise}}(c, t), \qquad \text{with } p_{\text{core}}(c, t) \equiv p_\star(c, t). \tag{13}$$

**Assumption 3** (Selector quality & independence within components). Let $Z(c, t) \in \{0, 1\}$ be the indicator that the token is kept by the filter. Define the error rates

$$\alpha := \Pr[Z = 0 \mid (c, t) \sim p_{\text{core}}], \qquad \beta := \Pr[Z = 1 \mid (c, t) \sim p_{\text{noise}}]. \tag{14}$$

We assume the selector has non–trivial skill: $\alpha + \beta < 1$. We consider two flavors of conditional independence:

- *Strong (MAR within components).* Given the latent component label $G \in \{\text{core}, \text{noise}\}$, $Z$ is independent of $(c, t)$:
$$Z \perp (c, t) \mid G.$$

- *Weak (bounded selection bias).* There exist $\rho_{\text{c}}, \rho_{\text{n}} \geq 0$ such that
$$\big\| \mathbb{E}[\phi_\theta(c, t) \mid G{=}\text{core}, Z{=}1] - \mathbb{E}[\phi_\theta(c, t) \mid G{=}\text{core}] \big\|_{M^{-1}} \leq \rho_{\text{c}} \big\| g_{\text{core}}(\theta) \big\|_{M^{-1}}, \tag{15}$$
$$\big\| \mathbb{E}[\phi_\theta(c, t) \mid G{=}\text{noise}, Z{=}1] - \mathbb{E}[\phi_\theta(c, t) \mid G{=}\text{noise}] \big\|_{M^{-1}} \leq \rho_{\text{n}} \big\| g_{\text{core}}(\theta) \big\|_{M^{-1}}. \tag{16}$$

**Assumption 4** (Teacher forcing invariance). Filtering removes token losses but does not change the context law $Q(c)$ in the forward pass. Consequently, $F_\lambda(\theta)$ in equation 10 is unaffected by filtering.

**Assumption 5** (Local smoothness of $\mathcal{L}_\star$). For each $\theta$ there exist $r > 0$ and $L = L(\theta, r) > 0$ such that, for all $u$ with $\|u\|_2 \le r$,

$$\mathcal{L}_\star(\theta + u) \le \mathcal{L}_\star(\theta) + \nabla \mathcal{L}_\star(\theta)^\top u + \frac{L}{2} \|u\|_2^2. \tag{17}$$

**Assumption 6** (Score–norm control for high–confidence tokens). Assume the logits $z_\theta(c) \in \mathbb{R}^K$ are coordinatewise $L_z$–Lipschitz in $\theta$, i.e. $\|\nabla_\theta z_\theta(c)_k\|_2 \le L_z$ for all $k$, uniformly in $c$. If $F_\lambda(\theta) \succeq \mu I$ for some $\mu > 0$, then for all $(c, t)$

$$\left\| \phi_\theta(c, t) \right\|_{F_\lambda^{-1}} \le \frac{2L_z}{\sqrt{\mu}} \left(1 - p_\theta(t \mid c)\right). \tag{18}$$

**Assumption 7** (Weak incoherence of noise). Let

$$g_{\text{core}}(\theta) := \mathbb{E}_{(c,t) \sim p_{\text{core}}}[\phi_\theta(c, t)], \qquad g_{\text{noise}}(\theta) := \mathbb{E}_{(c,t) \sim p_{\text{noise}}}[\phi_\theta(c, t)].$$

There exists $\zeta_M \in [0, 1)$ such that for all $\theta$

$$\langle g_{\text{core}}(\theta), g_{\text{noise}}(\theta) \rangle_{M^{-1}} \le \zeta_M \left\| g_{\text{core}}(\theta) \right\|_{M^{-1}}^2. \tag{19}$$

**Remark 1** (Proof of Assumption 6). For softmax $p_\theta(t \mid c) = \text{softmax}(z_\theta(c))_t$, the score decomposes as $\phi_\theta(c, t) = \sum_{k=1}^K (\mathbb{I}\{k = t\} - p_\theta(k \mid c)) \nabla_\theta z_\theta(c)_k$. By triangle inequality and $\|\nabla_\theta z_\theta(c)_k\|_2 \le L_z$,

$$\left\| \phi_\theta(c, t) \right\|_2 \le L_z \sum_{k=1}^K |\mathbb{I}\{k = t\} - p_\theta(k \mid c)| = 2L_z \left(1 - p_\theta(t \mid c)\right).$$

If $F_\lambda(\theta) \succeq \mu I$, then $\|v\|_{F_\lambda^{-1}} \le \mu^{-1/2} \|v\|_2$ for all $v$, which yields equation 18.

### A.3 DISTRIBUTIONS USED BY SGD AND GRADIENT IDENTITIES

Let D be a distribution on $(c, t)$; write

$$g(\theta; \mathsf{D}) := \mathbb{E}_{(c,t) \sim \mathsf{D}}[\phi_\theta(c, t)].$$

We denote by $p_{\text{train}}$ the unfiltered training distribution equation 13, and by $p_{\text{fil}}$ the filtered training distribution induced by keeping only tokens with $Z = 1$ and renormalizing.

**Lemma 1** (Filtering–induced mixture and gradient identities). Let $a := 1 - \varepsilon$, $b := \varepsilon$ and

$$Z_{\text{fil}} := a(1 - \alpha) + b\beta. \tag{20}$$

Then

$$g(\theta; p_{\text{train}}) = a\, g_{\text{core}}(\theta) + b\, g_{\text{noise}}(\theta), \tag{21}$$

$$(\text{Strong MAR}): \quad g(\theta; p_{\text{fil}}) = \frac{a(1 - \alpha)\, g_{\text{core}}(\theta) + b\beta\, g_{\text{noise}}(\theta)}{Z_{\text{fil}}}, \tag{22}$$

$$(\text{Weak bias}): \quad g(\theta; p_{\text{fil}}) = \frac{a(1 - \alpha)\, g_{\text{core}}^{\text{sel}}(\theta) + b\beta\, g_{\text{noise}}^{\text{sel}}(\theta)}{Z_{\text{fil}}}, \tag{23}$$

where $g_{\text{core}}^{\text{sel}} := \mathbb{E}[\phi_\theta \mid G = \text{core}, Z = 1]$ and $g_{\text{noise}}^{\text{sel}}$ is defined analogously. Moreover,

$$\nabla_\theta \mathcal{L}_\star(\theta) = -g_{\text{core}}(\theta). \tag{24}$$

*Proof.* equation 21 is linearity of expectation applied to equation 13. For equation 22, under $Z \perp (c, t) \mid G$,

$$\mathbb{E}\big[\phi_\theta(c, t)\, \mathbb{I}\{Z = 1\} \mid G = \text{core}\big] = (1 - \alpha)\, \mathbb{E}[\phi_\theta(c, t) \mid G = \text{core}] = (1 - \alpha)\, g_{\text{core}},$$

and similarly for noise; after renormalization by $Z_{\text{fil}}$ we obtain equation 22. If independence is replaced by the weak bias bounds equation 15–equation 16, the same computation yields equation 23. Finally, equation 24 is the usual score–matching identity since $p_\star$ does not depend on $\theta$. $\qquad\square$

A.4 ALIGNMENT AND ITS IMPROVEMENT UNDER FILTERING (GENERAL SPD $M$)

Given an SPD $M(\theta) \succ 0$, define the $M$–*alignment* of any preconditioned direction $\widetilde{g}$ with the ideal direction by

$$\mathcal{A}_M(\theta; \widetilde{g}) := \big\langle \widetilde{\nabla}_M \mathcal{L}_\star(\theta), \, \widetilde{g} \big\rangle_M. \qquad (25)$$

In particular, if we use the $M$–preconditioned gradient of a distribution D, i.e. $\widetilde{g}(\theta; \mathsf{D}) := -M(\theta)^{-1} g(\theta; \mathsf{D})$, then by equation 24 and equation 25

$$\mathcal{A}_M(\theta; \widetilde{g}(\cdot; \mathsf{D})) = \big\langle - M^{-1} g_{\mathrm{core}}, \, -M^{-1} g(\cdot; \mathsf{D}) \big\rangle_M = g_{\mathrm{core}}^\top M^{-1} g(\theta; \mathsf{D}). \qquad (26)$$

**Lemma 2** (Explicit alignments under unfiltered and filtered sampling). Under Assumptions 2–4, for $a = 1 - \varepsilon$, $b = \varepsilon$, and $Z_{\mathrm{fil}}$ as in equation 20, we have

$$\mathcal{A}_M^{\mathrm{train}}(\theta) := \mathcal{A}_M\big(\theta; \widetilde{g}(\cdot; p_{\mathrm{train}})\big) = a \, \|g_{\mathrm{core}}\|_{M^{-1}}^2 \, + \, b \, \langle g_{\mathrm{core}}, g_{\mathrm{noise}} \rangle_{M^{-1}}, \quad (27)$$

$$\text{(Strong MAR):} \quad \mathcal{A}_M^{\mathrm{fil}}(\theta) := \mathcal{A}_M\big(\theta; \widetilde{g}(\cdot; p_{\mathrm{fil}})\big) = \frac{a(1 - \alpha)}{Z_{\mathrm{fil}}} \, \|g_{\mathrm{core}}\|_{M^{-1}}^2 \, + \, \frac{b\beta}{Z_{\mathrm{fil}}} \, \langle g_{\mathrm{core}}, g_{\mathrm{noise}} \rangle_{M^{-1}}. \qquad (28)$$

Under the weak–bias alternative in Assumption 3, the same formulas hold with the replacements $g_{\mathrm{core}} \mapsto g_{\mathrm{core}}^{\mathrm{sel}}$ and $g_{\mathrm{noise}} \mapsto g_{\mathrm{noise}}^{\mathrm{sel}}$ in the numerators.

*Proof.* Plug equation 21 and equation 22 into equation 26. □

**Theorem 1** (Filtering strictly improves $M$–alignment (strong MAR case)). Under Assumptions 2–4 and 7, with $a = 1 - \varepsilon$, $b = \varepsilon$, $Z_{\mathrm{fil}} = a(1 - \alpha) + b\beta$, the improvement in alignment is

$$\mathcal{A}_M^{\mathrm{fil}}(\theta) - \mathcal{A}_M^{\mathrm{train}}(\theta) = \frac{ab \, (1 - \alpha - \beta)}{Z_{\mathrm{fil}}} \Big( \|g_{\mathrm{core}}(\theta)\|_{M^{-1}}^2 \, - \, \langle g_{\mathrm{core}}(\theta), g_{\mathrm{noise}}(\theta) \rangle_{M^{-1}} \Big). \qquad (29)$$

In particular, if $\alpha + \beta < 1$ and equation 19 holds with $\zeta_M < 1$, then

$$\mathcal{A}_M^{\mathrm{fil}}(\theta) - \mathcal{A}_M^{\mathrm{train}}(\theta) \geq \frac{ab \, (1 - \alpha - \beta) \, (1 - \zeta_M)}{Z_{\mathrm{fil}}} \, \big\|g_{\mathrm{core}}(\theta)\big\|_{M^{-1}}^2 \, > \, 0. \qquad (30)$$

*Proof.* Subtract equation 27 from equation 28 and note

$$\frac{a(1 - \alpha)}{Z_{\mathrm{fil}}} - a = \frac{ab(1 - \alpha - \beta)}{Z_{\mathrm{fil}}} - \frac{b\beta}{Z_{\mathrm{fil}}} + b, \quad \frac{b\beta}{Z_{\mathrm{fil}}} - b = -\frac{ab(1 - \alpha - \beta)}{Z_{\mathrm{fil}}}.$$

This simplifies the difference to equation 29. The lower bound equation 30 follows from equation 19. □

**Theorem 2** (Robust alignment gain under bounded selection bias). Under Assumptions 2–4, 7, and the weak–bias bounds equation 15–equation 16, the alignment improvement satisfies

$$\mathcal{A}_M^{\mathrm{fil}}(\theta) - \mathcal{A}_M^{\mathrm{train}}(\theta) \geq \frac{ab \, (1 - \alpha - \beta) \, (1 - \zeta_M)}{Z_{\mathrm{fil}}} \, \big\|g_{\mathrm{core}}\big\|_{M^{-1}}^2 - \frac{a(1 - \alpha)\rho_{\mathrm{c}} + b\beta\rho_{\mathrm{n}}}{Z_{\mathrm{fil}}} \, \big\|g_{\mathrm{core}}\big\|_{M^{-1}}^2. \qquad (31)$$

Hence the net gain remains positive whenever

$$ab(1 - \alpha - \beta)(1 - \zeta_M) > a(1 - \alpha)\rho_{\mathrm{c}} + b\beta\rho_{\mathrm{n}}.$$

*Proof.* Repeat the proof of Theorem 1 using equation 23. Add and subtract the strong–MAR numerators, then apply equation 15–equation 16 and Cauchy–Schwarz in $\langle \cdot, \cdot \rangle_{M^{-1}}$ to bound the deviation terms by $\rho_{\mathrm{c}} \|g_{\mathrm{core}}\|_{M^{-1}}^2$ and $\rho_{\mathrm{n}} \|g_{\mathrm{core}}\|_{M^{-1}}^2$, respectively. □

**Remark 2** (Edge cases). If $\varepsilon = 0$ (no noise) or $\alpha + \beta = 1$ (random selector), then equation 29 yields $\mathcal{A}_M^{\mathrm{fil}} = \mathcal{A}_M^{\mathrm{train}}$, i.e. no gain; if $\zeta_M = 1$ (perfect coherent noise), the lower bound equation 30 is zero.

## A.5 ONE–STEP DECREASE IN $\mathcal{L}_\star$: GENERAL PRECONDITIONERS AND EUCLIDEAN SGD

**Lemma 3** (One–step decrease for $M$–preconditioned steps (local version)). Let $\widetilde{g}$ be any direction and consider the update $\theta^+ = \theta - \eta\,\widetilde{g}$. Under Assumption 5 with radius $r > 0$, if $\eta\|\widetilde{g}\|_2 \leq r$, then

$$\mathcal{L}_\star(\theta^+) \ \leq \ \mathcal{L}_\star(\theta) \ - \ \eta\,\langle\widetilde{\nabla}_M\mathcal{L}_\star(\theta),\,\widetilde{g}\rangle_M \ + \ \tfrac{L}{2}\,\eta^2\,\|\widetilde{g}\|_2^2. \tag{32}$$

In particular, for $\widetilde{g}(\cdot;\mathsf{D}) = -M^{-1}g(\cdot;\mathsf{D})$ we have

$$\mathcal{L}_\star(\theta^+) \ \leq \ \mathcal{L}_\star(\theta) \ - \ \eta\,\mathcal{A}_M\big(\theta;\widetilde{g}(\cdot;\mathsf{D})\big) \ + \ \tfrac{L}{2}\,\eta^2\,\|\widetilde{g}(\cdot;\mathsf{D})\|_2^2.$$

*Proof.* By equation 17,

$$\mathcal{L}_\star(\theta^+) \leq \mathcal{L}_\star(\theta) + \nabla\mathcal{L}_\star(\theta)^\top(\theta^+ - \theta) + \tfrac{L}{2}\|\theta^+ - \theta\|_2^2.$$

With $\theta^+ - \theta = -\eta\widetilde{g}$ and $\widetilde{\nabla}_M\mathcal{L}_\star = M^{-1}\nabla\mathcal{L}_\star$, we obtain $\nabla\mathcal{L}_\star^\top(-\eta\widetilde{g}) = -\eta\,\nabla\mathcal{L}_\star^\top MM^{-1}\widetilde{g} = -\eta\,\langle\widetilde{\nabla}_M\mathcal{L}_\star,\widetilde{g}\rangle_M$. $\square$

**Corollary 1** (Filtered vs. unfiltered one–step improvement). Consider the two updates with common step size $\eta > 0$:

$$\theta^+_{\mathrm{fil}} = \theta - \eta\,\widetilde{g}(\cdot;p_{\mathrm{fil}}), \qquad \theta^+_{\mathrm{train}} = \theta - \eta\,\widetilde{g}(\cdot;p_{\mathrm{train}}).$$

If $\eta\max\{\|\widetilde{g}(\cdot;p_{\mathrm{fil}})\|_2, \|\widetilde{g}(\cdot;p_{\mathrm{train}})\|_2\} \leq r$, then

$$\mathcal{L}_\star(\theta^+_{\mathrm{fil}}) - \mathcal{L}_\star(\theta^+_{\mathrm{train}}) \ \leq \ -\eta\Big(\mathcal{A}_M^{\mathrm{fil}}(\theta) - \mathcal{A}_M^{\mathrm{train}}(\theta)\Big) \ + \ \frac{L}{2}\,\eta^2\Big(\|\widetilde{g}(\cdot;p_{\mathrm{fil}})\|_2^2 - \|\widetilde{g}(\cdot;p_{\mathrm{train}})\|_2^2\Big). \tag{33}$$

In particular, whenever

$$0 < \eta \leq \min\left\{\frac{2\big(\mathcal{A}_M^{\mathrm{fil}}(\theta) - \mathcal{A}_M^{\mathrm{train}}(\theta)\big)}{L\big(\|\widetilde{g}(\cdot;p_{\mathrm{fil}})\|_2^2 + \|\widetilde{g}(\cdot;p_{\mathrm{train}})\|_2^2\big)}\,,\ \frac{r}{\max\{\|\widetilde{g}(\cdot;p_{\mathrm{fil}})\|_2, \|\widetilde{g}(\cdot;p_{\mathrm{train}})\|_2\}}\right\}, \tag{34}$$

we have $\mathcal{L}_\star(\theta^+_{\mathrm{fil}}) \leq \mathcal{L}_\star(\theta^+_{\mathrm{train}})$, with strict inequality if the alignment gain in equation 30 (or equation 31) is strict.

**Remark 3** (Euclidean SGD and Adam). Lemma 3 and Corollary 1 hold for *any* SPD $M$; taking $M = I$ yields the standard Euclidean form

$$\mathcal{L}_\star(\theta - \eta g) \leq \mathcal{L}_\star(\theta) - \eta\langle\nabla\mathcal{L}_\star, g\rangle + \tfrac{L}{2}\eta^2\|g\|_2^2$$

and the same alignment improvement conclusions with $M^{-1}$–inner products replaced by Euclidean inner products. Taking $M = F_\lambda$ recovers the damped natural gradient geometry.

## A.6 HIGH–CONFIDENCE (KN) TOKENS: NEGLIGIBLE ALIGNMENT CONTRIBUTION

We show that high–confidence tokens (KN) have vanishing contribution to alignment in the Fisher geometry, with a *quantitative* bound relying on Assumption 6.

**Lemma 4** (KN tokens have vanishing Fisher contribution). Let $\mathcal{S}_{\mathrm{KN}} \subseteq \mathrm{Supp}(p_{\mathrm{train}})$ be any measurable set such that $p_\theta(t \mid c) \geq 1 - \delta$ for all $(c,t) \in \mathcal{S}_{\mathrm{KN}}$, with $\delta \in (0,1)$. Then, under Assumption 6,

$$\Big\|\mathbb{E}_{(c,t)\sim p_{\mathrm{train}}}\big[\phi_\theta(c,t)\,\mathbb{I}\{(c,t) \in \mathcal{S}_{\mathrm{KN}}\}\big]\Big\|_{F_\lambda^{-1}} \ \leq \ \frac{2L_z}{\sqrt{\mu}}\,\delta\,p_{\mathrm{train}}(\mathcal{S}_{\mathrm{KN}}). \tag{35}$$

Consequently, using Cauchy–Schwarz in $\langle\cdot,\cdot\rangle_{F_\lambda^{-1}}$,

$$\Big|\mathcal{A}_{F_\lambda}\big(\theta;\widetilde{g}(\cdot;p_{\mathrm{train}})\big) - \mathcal{A}_{F_\lambda}\big(\theta;\widetilde{g}(\cdot;p_{\mathrm{train}} \setminus \mathcal{S}_{\mathrm{KN}})\big)\Big| \ \leq \ \frac{2L_z}{\sqrt{\mu}}\,\delta\,p_{\mathrm{train}}(\mathcal{S}_{\mathrm{KN}})\,\big\|g_{\mathrm{core}}(\theta)\big\|_{F_\lambda^{-1}}. \tag{36}$$

*Proof.* By Jensen and equation 18,

$$\Big\|\mathbb{E}[\phi\,\mathbb{I}_{\mathcal{S}_{\mathrm{KN}}}]\Big\|_{F_\lambda^{-1}} \ \leq \ \mathbb{E}\big[\|\phi\|_{F_\lambda^{-1}}\mathbb{I}_{\mathcal{S}_{\mathrm{KN}}}\big] \ \leq \ \frac{2L_z}{\sqrt{\mu}}\,\mathbb{E}\big[(1 - p_\theta(t \mid c))\mathbb{I}_{\mathcal{S}_{\mathrm{KN}}}\big] \ \leq \ \frac{2L_z}{\sqrt{\mu}}\,\delta\,p_{\mathrm{train}}(\mathcal{S}_{\mathrm{KN}}).$$

The alignment bound follows from equation 26 and Cauchy–Schwarz. $\square$

**Remark 4** (From Fisher to general $M$). If $M \succ 0$ is any SPD matrix, then for all $v$, $\|v\|_{M^{-1}} \leq \|M^{-1/2}F_\lambda^{1/2}\|_{\mathrm{op}} \cdot \|v\|_{F_\lambda^{-1}}$. Thus Lemma 4 transfers to $\langle\cdot,\cdot\rangle_{M^{-1}}$ up to the condition–number factor $\|M^{-1/2}F_\lambda^{1/2}\|_{\mathrm{op}}$.

## A.7 COMBINING TR/RI WITH KN

Let $\mathcal{S}_{\text{TR}\downarrow} \cup \mathcal{S}_{\text{RI}\downarrow}$ denote the subset labeled *low relevance/importance* by the selector (mass $\varepsilon$ under $p_{\text{train}}$), and let $\mathcal{S}_{\text{KN}\downarrow}$ denote the subset of *high–confidence* tokens with $p_\theta(t \mid c) \geq 1 - \delta$.

Combining Theorems 1–2 with Lemma 4 (for $M = F_\lambda$), the net alignment improvement from filtering $\mathcal{S}_{\text{TR}\downarrow} \cup \mathcal{S}_{\text{RI}\downarrow} \cup \mathcal{S}_{\text{KN}\downarrow}$ satisfies

$$
\mathcal{A}_{F_\lambda}^{\text{fil}}(\theta) - \mathcal{A}_{F_\lambda}^{\text{train}}(\theta) \geq \frac{(1 - \varepsilon)\varepsilon(1 - \alpha - \beta)(1 - \zeta_{F_\lambda})}{Z_{\text{fil}}} \|g_{\text{core}}(\theta)\|_{F_\lambda^{-1}}^2
$$
$$
- \frac{2L_z}{\sqrt{\mu}} \delta \, p_{\text{train}}(\mathcal{S}_{\text{KN}\downarrow}) \|g_{\text{core}}(\theta)\|_{F_\lambda^{-1}}
$$
$$
- \frac{(1 - \varepsilon)(1 - \alpha)\rho_{\text{c}} + \varepsilon\beta\rho_{\text{n}}}{Z_{\text{fil}}} \|g_{\text{core}}(\theta)\|_{F_\lambda^{-1}}^2 \qquad (37)
$$

where the last (bias) term is absent under strong MAR. For sufficiently small $\delta$ and small selection bias $(\rho_{\text{c}}, \rho_{\text{n}})$, the positive TR/RI term dominates and the one–step KL improvement of the filtered update is strictly larger (Corollary 1).

## A.8 CONSEQUENCES FOR THE MODIFIED LOSS $\mathcal{L}_{\text{F}}$

Under teacher forcing (Assumption 4), excluding tokens identified by the selector from the token sum exactly replaces the expectation with respect to $p_{\text{train}}$ by the renormalized expectation with respect to $p_{\text{fil}}$ in the expression for the stochastic gradient (or its $M$–preconditioned variant). Therefore Theorems 1–2 and Corollary 1 apply directly to the SGD (or $M$–preconditioned SGD) dynamics for $\mathcal{L}_{\text{F}}$.

## A.9 ADDITIONAL REMARKS AND CAVEATS

1. **Estimator independence & selection bias.** Equation equation 22 relies on the strong MAR–within–component assumption in Assumption 3. When MAR fails, equation 23 and Theorem 2 quantify the degradation through $(\rho_{\text{c}}, \rho_{\text{n}})$.

2. **Geometry and optimizers.** All alignment statements are given for an *arbitrary* SPD preconditioner $M$. The Fisher choice $M = F_\lambda$ confers parameterization invariance (up to damping), while $M = I$ matches Euclidean SGD; Adam/K–FAC correspond to other choices of $M$. Corollary 1 therefore bridges theory and practice without further assumptions.

3. **Smoothness is local.** We use local smoothness (Assumption 5) within a radius $r$ around $\theta$ to control the second–order term. The step–size condition equation 34 ensures $\theta^+$ remains in this local region.

4. **On variance claims.** In general, filtering with *renormalization* does not guarantee a universal reduction in gradient covariance; the effect is data– and selector–dependent. Our theoretical results deliberately refrain from claiming such a reduction. Empirically, we observe reduced gradient dispersion when removing KN tokens; a rigorous sufficient condition (e.g. stratified second–moment dominance) leads to provable variance reduction, but this is orthogonal to the alignment results developed here.

5. **Parameter estimation.** The selector ROC $(\alpha, \beta)$ and the mixture mass $\varepsilon$ can be estimated on a small annotated split; $\zeta_M$ can be estimated from held–out data via $\langle g_{\text{core}}, g_{\text{noise}}\rangle_{M^{-1}}/\|g_{\text{core}}\|_{M^{-1}}^2$. These estimators make the alignment gain equation 29 empirically checkable.

6. **Notation.** We write $Z_{\text{fil}}$ for the filtering normalizer equation 20 to avoid confusion with the Fisher matrix $F_\lambda$.

## A.10 SELF–CONTAINED PROOFS OF KEY AUXILIARY FACTS

**Lemma 5** (Ideal gradient equals negative core expectation)**.** Under teacher forcing with $p_{\text{core}} \equiv p_\star$, we have $\nabla_\theta \mathcal{L}_\star(\theta) = - g_{\text{core}}(\theta)$.

*Proof.* By definition, $\mathcal{L}_\star(\theta) = \mathbb{E}_{(c,t)\sim p_\star}[-\log p_\theta(t \mid c)]$, so $\nabla_\theta \mathcal{L}_\star(\theta) = -\mathbb{E}_{(c,t)\sim p_\star}[\nabla_\theta \log p_\theta(t \mid c)] = -g_{\text{core}}(\theta)$. $\qquad\square$

**Lemma 6** (Score–norm bound in Fisher geometry). *Under Assumption 6, for all $(c,t)$,* $\|\phi_\theta(c,t)\|_{F_\lambda^{-1}} \leq \frac{2L_z}{\sqrt{\mu}}(1 - p_\theta(t \mid c))$.

*Proof.* Shown in Remark 1. $\qquad\square$

**Lemma 7** (Alignment identity). *For any SPD $M \succ 0$,* $\langle \widetilde{\nabla}_M \mathcal{L}_\star, -M^{-1}g(\cdot; \mathsf{D})\rangle_M = g_{\text{core}}^\top M^{-1} g(\cdot; \mathsf{D})$.

*Proof.* $\langle M^{-1}\nabla \mathcal{L}_\star, -M^{-1}g\rangle_M = -\nabla \mathcal{L}_\star^\top M^{-1}g = g_{\text{core}}^\top M^{-1}g$ by Lemma 5. $\qquad\square$

SUMMARY OF THE STRENGTHENED GUARANTEE

Under Assumptions 1–7, token filtering induces a renormalized training distribution $p_{\text{fil}}$ such that, for any SPD preconditioner $M$,

$$\mathcal{A}_M^{\text{fil}}(\theta) - \mathcal{A}_M^{\text{train}}(\theta) = \frac{(1-\varepsilon)\varepsilon(1-\alpha-\beta)}{Z_{\text{fil}}}\Big(\|g_{\text{core}}\|_{M^{-1}}^2 - \langle g_{\text{core}}, g_{\text{noise}}\rangle_{M^{-1}}\Big),$$

and hence, under the incoherence condition $\zeta_M < 1$, the alignment gain is *strictly positive* and explicitly lower–bounded by equation 30. By the local one–step bound (Corollary 1), for sufficiently small step sizes, a single filtered update yields a strictly larger *decrease* of the ideal risk $\mathcal{L}_\star$ than the unfiltered update. Finally, KN tokens (high confidence under the base model) have vanishing Fisher contribution with an $O(\delta)$ bound (Lemma 4), so removing them incurs negligible alignment cost while empirically stabilizing optimization. All statements specialize to the Fisher geometry ($M = F_\lambda$) for parameterization–invariant conclusions and to $M = I$ for Euclidean SGD/Adam–style updates.

# B MORE DETAILS

## B.1 OTSU METHOD

**Formulation.** For a histogram with gray levels in the range $[0, L-1]$, let threshold $t$ divide the pixels into two classes $C_0$ (0 to $t$) and $C_1$ ($t+1$ to $L-1$). The between-class variance $\sigma_b^2(t)$ is defined as:

$$\sigma_b^2(t) = \omega_0(t)\omega_1(t)\left[\mu_1(t) - \mu_0(t)\right]^2$$

where:

- $\omega_0(t) = \sum_{i=0}^{t} p(i)$, $\omega_1(t) = \sum_{i=t+1}^{L-1} p(i)$ are the class probability weights ($p(i)$ is the probability of gray level $i$)

- $\mu_0(t) = \frac{1}{\omega_0(t)}\sum_{i=0}^{t} ip(i)$, $\mu_1(t) = \frac{1}{\omega_1(t)}\sum_{i=t+1}^{L-1} ip(i)$ are the class means

The **optimal threshold** $t^*$ is determined by maximizing the between-class variance:

$$t^* = \arg\max_{0\leq t < L} \sigma_b^2(t)$$

**Multi-Otsu Extension.** For partitioning into $k$ classes requiring $k-1$ thresholds $t_1, t_2, \ldots, t_{k-1}$, the objective function becomes:

$$\sigma_b^2(t_1, \ldots, t_{k-1}) = \sum_{m=0}^{k-1} \omega_m \left(\mu_m - \mu_T\right)^2$$

where $\mu_T$ is the global mean. The optimal threshold combination is obtained by maximizing the total between-class variance.

## B.2 BASELINES SETTING

Here we describe all baseline methods in detail and explain their roles in our experiment.

1) **Regular LLM Implementations**:

- **CA**: This represents the performance of the original base model, ensuring that our fine-tuning is correct and effective.

- **Normal**: This gives the performance of regular fine-tuning, serving as a direct reference to observe the influence of not applying token-level noise filtering.

- **More Epochs ($\times$2 Epochs)**: This method uses double the training epochs for fine-tuning, eliminating bias caused by recording experimental results before complete convergence.

2) **Data Enhancement Methods**: Since our `XTF` is essentially a dataset enhancement method that optimizes fine-tuning performance before training the model, we adopt two mainstream data enhancement methods as baselines: data augmentation (DA) and data filtering (DF).

- **DA**: We use the state-of-the-art data augmentation method, which leverages high-performance LLMs to optimize the dataset (Dai et al., 2025). Specifically, we employ Claude-3.5-sonnet (Anthropic, 2025) in our experiment.

- **DF**: Recent studies have shown that removing data items with high perplexity can improve training performance (Li et al., 2019; Ankner et al., 2024). Thus, we filter out the top-5% perplexity data items in this baseline experiment.

- **SLM**: This is a novel method that can effectively filter noisy tokens in pre-training tasks. Vanilla SLM relies on a high-quality dataset to train a reference model, which is impractical in fine-tuning tasks. Therefore, we train the reference model using a subset of the fine-tuning dataset to simulate the effect of SLM as closely as possible.

- **TC**: This is a fine-grained token-level data selection method for LLM supervised fine-tuning, dedicated to filtering uninformative tokens while preserving task-specific informative ones—effectively addressing token-level noise even in high-quality samples (Pang et al., 2025). In our experiment, we adopt TC's Self-Evolving Cleaning strategy, with implementation tailored to the epoch-wise training process: Specifically, in each training epoch, we treat the model trained in the previous epoch as the reference model and the model to be updated in the current epoch as the base model. For every token in the training dataset, we calculate its quality score using the loss disparity between this base model and the reference model (a core scoring logic of Self-Evolving Cleaning).

For a fair experiment, we use the same validation dataset to select the best model for all baselines.

## B.3 HYPERPARAMETERS SETTING

We carefully adjust the hyperparameter to make sure our fine-tuning process is correct and effective, and we finally select a common setting shown in Table 3. Besides, we set `torch_dtype = bfloat16` and adopt default generation setting (*e.g.*, temperature) in all of our experiments. We conduct our experiment on NVIDIA A100-SMX-80G, H20-96G and L20-48G.

Table 3: Hyperparameters for different models and tasks. Ma1 denotes math (gsm8k);Ma2 denotes math(NuminaMath-CoT for MATH-500 evaluatio); Fi denotes finance; Co denotes code; Me denotes medicine. Lr denotes learning rate, Ep denotes epoch number of fine-tuning, and MNT denotes max new tokens for model evaluation.

| Model | Task | Data scale | Lr | Lr_LoRA | Ep | Ep_LoRA | MNT |
|-------|------|-----------|-----|---------|-----|---------|-----|
| 1B | Ma1 | 500 | 9e-6 | - | 3 | - | 512 |
| 1B | Ma2 | 500 | 9e-6 | - | 3 | - | 2048 |
| 1B | Fi | 500 | 9e-6 | - | 3 | - | 1024 |
| 1B | Co | 500 | 9e-6 | - | 3 | - | 512 |
| 1B | Me | 500 | 1e-5 | - | 3 | - | 1024 |
| 1.5B | Ma1 | 600 | 9e-6 | - | 3 | - | 512 |
| 1.5B | Ma2 | 600 | 9e-6 | - | 3 | - | 2048 |
| 1.5B | Fi | 600 | 9e-6 | - | 3 | - | 1024 |
| 1.5B | Co | 600 | 9e-6 | - | 3 | - | 512 |
| 1.5B | Me | 600 | 1e-5 | - | 3 | - | 1024 |
| 3B | Ma1 | 1000 | 8e-6 | - | 3 | - | 512 |
| 3B | Ma2 | 1000 | 8e-6 | - | 3 | - | 2048 |
| 3B | Fi | 1000 | 8e-6 | - | 3 | - | 1024 |
| 3B | Co | 1000 | 8e-6 | - | 3 | - | 512 |
| 3B | Me | 1000 | 3e-6 | - | 3 | - | 1024 |
| 7B&8B | Ma1 | 3000 | 5e-/6 | 5e-5 | 3 | 5 | 512 |
| 7B&8B | Ma2 | 3000 | 5e-6 | 5e-5 | 3 | 5 | 2048 |
| 7B&8B | Fi | 3000 | 5e-6 | 5e-5 | 3 | 5 | 1024 |
| 7B&8B | Co | 3000 | 5e-6 | 5e-5 | 3 | 5 | 512 |
| 7B&8B | Me | 3000 | 2e-6 | 5e-5 | 3 | 5 | 1024 |
| 14B | Ma1 | 3000 | 3e-6 | 3e-5 | – | 5 | 512 |
| 14B | Ma2 | 3000 | 3e-6 | 3e-5 | – | 5 | 2048 |
| 14B | Fi | 3000 | 3e-6 | 3e-5 | – | 5 | 1024 |
| 14B | Co | 3000 | 3e-6 | 3e-5 | – | 5 | 512 |
| 14B | Me | 3000 | 3e-6 | 3e-5 | – | 5 | 1024 |

## C    MORE EXPERIMENT RESULTS

### C.1    EXTENSION OF MAIN EXPERIMENT

Two experimental setups were implemented: (1) Models were fine-tuned on NuminaMath-CoT and subsequently evaluated on MATH-500; (2) Models were fine-tuned and evaluated end-to-end on the FIQA dataset. For space considerations, additional experimental results from one economics dataset are included in the appendix, where all configurations are detailed in Table 3.

The result are shown in Table 4. In math task, XTF has average 4.6% higher accuracy than normal fine-tuning and 1.3% higher accuracy than the best baseline SLM. In finance task, XTF has average 4.8% higher accuracy than normal fine-tuning and 2.5% higher accuracy than the best baseline TC. The two tasks are more complex than the gsm8k and medicine tasks used in Section 4.2, and the final model accuracies are generally lower, especially for small-scale models. Therefore, the average advantage achieved by XTF is not as pronounced as in Section 4.2. However, XTF still achieves the best performance in 16 out of 20 cases and the second-best performance in 2 cases, which is sufficient to demonstrate the reliable advantage of the XTF method.

### C.2    ABLATION STUDY ABOUT THRESHOLD

In the threshold ablation study, we explore how the three thresholds influence the percentage of filtered tokens and the final training result. This ablation study is conducted using Deepseek-distilled-qwen-1.5B on the GSM8K dataset.

Table 4: Extension of Main Experiment. We show the accuracy of LLMs across different fine-tuning methods. Best results are marked in **bold** and the second best results are marked with underline.

| MATH:Fine-tuning models on NuminaMath-CoT and Evaluate models on MATH-500 | | | | | | | | | | |
|---|---|---|---|---|---|---|---|---|---|---|
| **Model** | $|\theta|$ | **LoRA** | **CA** | **Normal** | **×2 Ep** | **DF** | **DA** | **SLM** | **TC** | **XTF** |
| Llama-3.2 | 1B | × | 0.0 | 2.8 | 3.0 | 1.5 | 2.1 | 3.0 | 2.9 | **3.6** |
| Llama-3.2 | 3B | × | 0.8 | 4.8 | 4.9 | 5.2 | 4.9 | 8.2 | 8.4 | **8.6** |
| Llama-3.1 | 8B | × | 4.9 | 19.8 | 19.8 | **23.6** | 19.2 | 22.8 | 22.3 | 23.4 |
| Llama-3.1 | 8B | ✓ | 4.9 | 20.7 | 21.1 | 19.5 | 18.8 | 24.7 | 24.2 | **28.7** |
| Mistral | 7B | × | 12.3 | 26.6 | 26.9 | 26.4 | 25.8 | 27.3 | 27.1 | **30.8** |
| Mistral | 7B | ✓ | 12.3 | 26.4 | 26.8 | 29.1 | 27.3 | **33.4** | 33.2 | 32.5 |
| Deepseek-distilled-qwen | 1.5B | × | 33.4 | 50.6 | 50.8 | 52.3 | 49.2 | 51.2 | 50.5 | **52.6** |
| Deepseek-distilled-qwen | 7B | × | 44.5 | 65.3 | 65.9 | 55.7 | 61.4 | 69.2 | 69.5 | **70.3** |
| Deepseek-distilled-qwen | 7B | ✓ | 44.5 | 49.6 | 49.8 | 51.2 | 52.8 | 55.3 | 53.2 | **57.4** |
| Deepseek-distilled-qwen | 14B | ✓ | 52.8 | 72.1 | 72.1 | 63.4 | 68.9 | 75.3 | 74.6 | **77.2** |
| Average | – | – | 21.0 | 33.9 | 34.1 | 32.8 | 33.0 | 37.0 | 36.6 | **38.3** |
| **Fine-tuning and evaluate models on fiqa** | | | | | | | | | | |
| **Model** | $|\theta|$ | **LoRA** | **CA** | **Normal** | **×2 Ep** | **DF** | **DA** | **SLM** | **TC** | **XTF** |
| Llama-3.2 | 1B | × | 0.0 | 0.4 | 0.4 | **0.6** | 0.3 | 0.5 | 0.5 | 0.5 |
| Llama-3.2 | 3B | × | 0.5 | 1.1 | 1.0 | 2.2 | 1.6 | 3.2 | 3.4 | **3.5** |
| Llama-3.1 | 8B | × | 2.5 | 6.8 | 6.9 | 5.1 | 8.2 | 7.9 | 7.6 | **8.4** |
| Llama-3.1 | 8B | ✓ | 2.5 | 5.2 | 5.4 | 4.3 | 4.8 | **8.9** | 8.6 | 8.1 |
| Mistral | 7B | × | 8.7 | 13.3 | 13.3 | 12.4 | 14.1 | 15.1 | 14.8 | **18.5** |
| Mistral | 7B | ✓ | 8.7 | 12.8 | 12.9 | 13.6 | 12.9 | 14.2 | 14.9 | **17.2** |
| Deepseek-distilled-qwen | 1.5B | × | 11.9 | 18.4 | 18.7 | 15.7 | 19.2 | 21.5 | 20.8 | **25.9** |
| Deepseek-distilled-qwen | 7B | × | 18.4 | 28.5 | 28.9 | 28.3 | 30.9 | 31.6 | 31.9 | **36.1** |
| Deepseek-distilled-qwen | 7B | ✓ | 18.4 | 22.6 | 22.7 | 20.8 | 23.9 | 22.8 | 22.1 | **29.3** |
| Deepseek-distilled-qwen | 14B | ✓ | 26.3 | 35.8 | 36.2 | 31.5 | 34.1 | 40.7 | 41.2 | **45.6** |
| Average | – | – | 9.8 | 14.5 | 14.6 | 13.5 | 15 | 16.6 | 16.8 | **19.3** |

(a) RI Filtering     (b) KN Filtering     (c) TR Filtering

Figure 6: Filtered token percentage under different threshold disturbance. In the ablation study of RI and TR, the value 0 on the horizontal axis represents the original threshold without any perturbation. In the ablation study of KN, the horizontal axis corresponds to different PCP values.

**Filtered Tokens.** First, we investigate whether adjusting the XTF thresholds produces a noticeable effect on the total number of filtered tokens. Specifically, for RI and TR, we first apply normalization to the scores and then modify the thresholds obtained by XTF to compute the number of filtered tokens. For KN, we use different PCP values to perform token filtering. To evaluate the stability of the filtering effect, we compute the proportion of filtered tokens relative to all label tokens under four different training-set sizes.

The results are shown in Figure 6. Directly adjusting the threshold value produces a significant impact on the number of filtered tokens. For RI, a slight increase in the threshold causes a large number of tokens (>70%) to be filtered. For KN, each 1% change in PCP corresponds to approximately 2% of all tokens, which exceeds 15% of the number of tokens filtered under the 0.95 PCP threshold. For TR, both increasing and decreasing the threshold lead to substantial changes. It is noteworthy that the trends of threshold variation are highly similar across different training-sizes, which indicates that the threshold-selection scheme based on score distributions is resistant to dataset scale.

Table 5: The ablation study on the rationality of the thresholds is conducted as follows: +3%, -3%, and -0% correspond to adjustments in the number of tokens filtered under the respective attributes, based on the scores to decide whether a token should be filtered or not. The top five results are marked in the table.

| RI($\downarrow$) | -3% | -3% | -3% | -3% | -3% | -3% | -3% | -3% | -3% |
|---|---|---|---|---|---|---|---|---|---|
| KN($\downarrow$) | -3% | -3% | -3% | -0% | -0% | -0% | +3% | +3% | +3% |
| TR($\downarrow$) | -3% | -0% | +3% | -3% | -0% | +3% | -3% | -0% | +3% |
| ACC (%) | 51.2 | 48.3 | 53.2 | 50.7 | 51.7 | 47.8 | 53.7 | 47.3 | 45.3 |
| Order | – | – | 5 | – | – | – | 4 | – | – |
| | | | | | | | | | |
| RI($\downarrow$) | -0% | -0% | -0% | -0% | -0% | -0% | 0% | 0% | 0% |
| KN($\downarrow$) | -3% | -3% | -3% | -0% | -0% | -0% | +3% | +3% | +3% |
| TR($\downarrow$) | -3% | -0% | +3% | -3% | -0% | +3% | -3% | -0% | +3% |
| ACC (%) | 44.3 | 50.5 | 47.8 | 47.3 | 56.2 | 54.2 | 49.8 | 48.0 | 52.8 |
| Order | – | – | – | – | 1 | 3 | – | – | – |
| | | | | | | | | | |
| RI($\downarrow$) | +3% | +3% | +3% | +3% | +3% | +3% | +3% | +3% | +3% |
| KN($\downarrow$) | -3% | -3% | -3% | -0% | -0% | -0% | +3% | +3% | +3% |
| TR($\downarrow$) | -3% | -0% | +3% | -3% | -0% | +3% | -3% | -0% | +3% |
| ACC (%) | 48.8 | 47.8 | 47.3 | 48.3 | 50.3 | 50.3 | 48.8 | 47.3 | 55.7 |
| Order | – | – | – | – | – | – | – | – | 2 |

**Final Training Result.** Since directly adjusting the threshold produces a large impact on the number of filtered tokens, we adopt a more fine-grained filtering scheme, namely, perturbing the percentage of filtered tokens in each sentence directly according to the ranking of the scores. We varied the number of filtered tokens along different attribute dimensions by ±3% [1]. We systematically combined all possible threshold settings, resulting in a total of 27 cases.

As shown in Table 5, our thresholds, which are computed based on statistical principles, consistently lead to the best fine-tuning performance across all cases. While we observe that certain alternative threshold settings could also yield competitive results, these settings do not conform to consistent patterns, such as linear or normal distributions. Therefore, we believe that it remains challenging to derive a reliable and generalizable threshold computation method through post-hoc adjustment of our statistically derived thresholds.

## D COMPUTATIONAL OVERHEAD

Table 6: Memory Usage (peak/normal) for different models. Only the scoring process is considered.

| Model | Metric | Math | Code | Medicine |
|---|---|---|---|---|
| Llama 1B | Mb (peak/normal) | 5362.2 / 4746.3 | 20149.0 / 4746.3 | 7617.2 / 4746.3 |
| Llama 3B | Mb (peak/normal) | 13093.7 / 12288.7 | 32465.6 / 12288.7 | 16035.3 / 12288.7 |
| Llama 8B | Mb (peak/normal) | 31740.7 / 30665.0 | 60487.2 / 30665.0 | 36525.3 / 30665.0 |
| Deepspeek 1.5B | Mb (peak/normal) | 7509.1 / 6811.8 | 18715.8 / 6811.8 | 7617.2 / 6811.8 |
| Deepspeek 7B | Mb (peak/normal) | 30439.1 / 29225.3 | 54760.8 / 29225.3 | 34626.5 / 29225.3 |
| Deepspeek 14B | Mb (peak/normal) | 58941.7 / 56375.2 | 92438.5 / 56375.2 | 68705.6 / 56375.2 |

We evaluate the GPU memory usage and time consumption of XTF across 6 models from two mainstream LLM (Large Language Model) series: Llama and Deepseek. The experimental setting of this experiment is aligned with that of the main experiment in Table 1. Additionally, we provide basic information about the selected datasets.

---

[1]A perturbation range of 3% is appropriate for our setting because the number of filtered tokens in a sentence is relatively small. If a perturbation of 1 or 2% is used, the filtered tokens in many sentences remain completely unchanged. A value of 3% is therefore a reasonable choice based on the experimental setting and granularity.

Table 7: Time Usage for different models. Only the scoring process is considered.

| Model | Metric | Math | Code | Medicine |
|---|---|---|---|---|
| Llama 1B | Time (s) | 134.62 | 150.42 | 253.68 |
| Llama 3B | Time (s) | 324.95 | 355.31 | 608.85 |
| Llama 8B | Time (s) | 1167.23 | 1541.27 | 2637.92 |
| Deepspeek 1.5B | Time (s) | 191.44 | 191.32 | 377.20 |
| Deepspeek 7B | Time (s) | 1179.11 | 1445.44 | 2276.91 |
| Deepspeek 14B | Time (s) | 3910.83 | 4132.56 | 4235.85 |

The GPU memory consumption of `XTF` is at the inference-level, and remains close to the memory required for model loading (normal state), without causing a significant increase in peak memory usage. Specifically, the maximum sequence length of the Code dataset reaches 2806, which is much longer than that of the Math (529) and Medicine (1231) datasets; thus, the Code dataset exhibits a relatively higher memory peak. However, this peak is still acceptable—far below the memory usage of fine-tuning, which is generally four times that of the model loading memory usage.

Time consumption is mainly determined by the average sequence length of the dataset. The Medicine dataset has an average sequence length of 401, which is longer than that of the Math (181) and Code (236) datasets, resulting in the highest time cost. Notably, although increasing the model size leads to higher computational costs, the theoretical overhead of `XTF` remains significantly lower than that of the most competitive token-level baseline, i.e., SLM (Token-Level Supervised Language Model)—the latter requires training an additional reference model. Details of this experiment are provided in the appendix of the paper.

**Cost Analysis.** In data processing techniques that involve LLMs, the time and computational costs mainly arise from model inference and training, and the cost of training far exceeds that of inference. Therefore, the cost levels can be divided into two major categories: (1) training-level cost: the token-level baselines, including SLM and TC, require training a reference model to assist with filtering, and they need to perform one inference on the original model and one inference on the reference model to obtain token-level loss values before computing scores. (2) inference-level cost: two types of sample-level methods fall into this category. The DF baseline needs to compute the average perplexity for each sample, and DA requires multiple LLM inferences to perform data augmentation. Although our `XTF` method is a token-level method, it completes token filtering with only two inferences. The first inference produces the output logits and attention, from which the RI score, KN score, and the domain vector for TR are obtained. The second inference computes the TR score for each token. Arguably, existing token-level methods that rely on multiple rounds of training on a reference model cannot compete with `XTF` in terms of cost, and the cost of sample-level methods is also not substantially lower than that of `XTF`.

### D.1 DISTRIBUTION FIGURES

Here we present more detailed distribution plots of the three scores. As shown in Figure 7, although the score distributions differ across various models or datasets, they still generally align with the patterns discussed in Section 3. Specifically, the reasoning importance scores are often concentrated at certain fixed values, with only a few extreme values observed. The knowledge novelty score distribution is relatively uniform, making it difficult to partition. The task relevance score distribution exhibits distinct clustering characteristics, thus enabling effective partitioning via clustering methods.

## E LLM USAGE

An OpenAI LLM (GPT-4o) was utilized as an assistant for writing and formatting, specifically to refine and suggest edits to figure and table captions, including improvements in grammar, phrasing, clarity, and layout (e.g., column alignment, caption length, and placement). The model's role was strictly limited to surface-level text and visual edits, without contributing to research ideation, experimental design, data analysis, or technical content. All outputs were thoroughly reviewed and revised by the authors, who retain full responsibility for the final text and visuals.

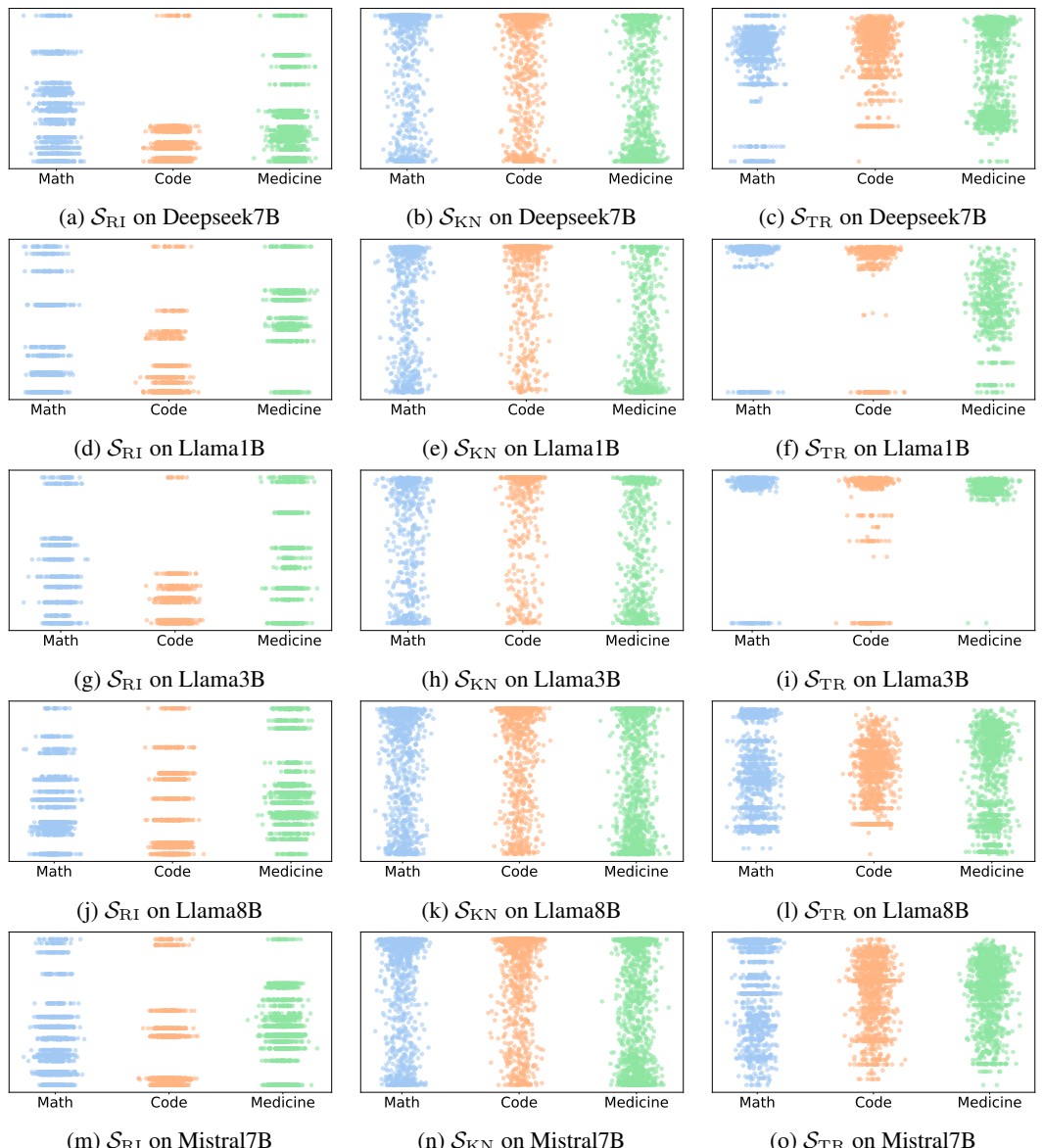

Figure 7: Distribution of three scores across different LLMs.

# F    EXAMPLES FOR TOKEN-LEVEL NOISE

Here we show some specific results of token-level noise filtering. As shown in Figure 8, Figure 9, Figure 10, Figure 11 and Figure 12, the part marked in yellow is the noisy tokens filtered out according to the corresponding scores.

We would like to point out that the filtering decisions made by XTF may, in some cases, be more reliable than human intuition. Take GSM8K as an example: tokens related to mathematical operations (e.g., "+", "=") or even specific computed results are not always necessary for effective fine-tuning. From the perspective of Knowledge Novelty (KN)—a dimension easier to interpret—we show in Figure 7 that the filtering prediction probability of the third "+" in the expression "2P + P + 4 = 3P + 4 = 220" exceeds 95%, and the token "216" in "220 - 4 = 216" is also filtered out. This outcome deviates from typical human intuition, indicating that learning reasoning logic is more critical for the model than memorizing symbols or specific results.

```
******************************************************************************
```
**Input I**: Trevor is a comic book illustrator. In the past three months, he has drawn 220 pages of the comic. The comic he illustrates releases a new issue once a month. The first and second months' issues were the same number of pages, but the third month's issue was four pages longer than either of the others. How many pages was the first issue of the comic?
```
******************************************************************************
```
**Low RI**: Let P be the number of pages of the first issue of the comic. The second issue is the same number of pages, so the first and second issues are P + P = 2P pages. The third issue is 4 pages longer, so it is P + 4 pages. All three issues are 2P + P + 4 = 3P + 4 = 220 pages long. Thus, 3P = 220 − 4 = 216 pages. Therefore, the first issue of the comic was P = 216 / 3 = <<216/3=72>>72 pages.

**Low KN**: Let P be the number of pages of the first issue of the comic. The second issue is the same number of pages, so the first and second issues are P + P = 2P pages. The third issue is 4 pages longer, so it is P + 4 pages. All three issues are 2P + P + 4 = 3P + 4 = 220 pages long. Thus, 3P = 220 − 4 = 216 pages. Therefore, the first issue of the comic was P = 216 / 3 = <<216/3=72>>72 pages.

**Low TR**: Let P be the number of pages of the first issue of the comic. The second issue is the same number of pages, so the first and second issues are P + P = 2P pages. The third issue is 4 pages longer, so it is P + 4 pages. All three issues are 2P + P + 4 = 3P + 4 = 220 pages long. Thus, 3P = 220 − 4 = 216 pages. Therefore, the first issue of the comic was P = 216 / 3 = <<216/3=72>>72 pages.

```
******************************************************************************
```
**Input II**: Don buys recyclable bottles in a small town. Shop A normally sells him 150 bottles, shop B sells him 180 bottles and Shop C sells him the rest. How many bottles does Don buy from Shop C if he is capable of buying only 550 bottles?
```
******************************************************************************
```
**Low RI**: Between Shop A and Shop B he buys 150 bottles + 180 bottles = <<150 + 180 = 330>>330 bottles. Thus he can buy 550 bottles − 330 bottles = <<550 − 330 = 220>>220 bottles from shop C.

**Low KN**: Between Shop A and Shop B he buys 150 bottles + 180 bottles = <<150 + 180 = 330>>330 bottles. Thus he can buy 550 bottles − 330 bottles = <<550 − 330 = 220>>220 bottles from shop C.

**Low TR**: Between Shop A and Shop B he buys 150 bottles + 180 bottles = <<150 + 180 = 330>>330 bottles. Thus he can buy 550 bottles − 330 bottles = <<550 − 330 = 220>>220 bottles from shop C.

Figure 8: Low score tokens in gsm8k dataset.

We do not claim that XTF's scoring mechanism is absolutely flawless. To mitigate the risk of mistakenly removing important tokens, we intentionally adopt conservative thresholds for all three scoring dimensions. This strategy reduces the likelihood of filtering out valuable tokens, enabling XTF to maintain consistently strong performance.

```
************************************************************************
```
**Input I**: Consider the terms of an arithmetic sequence: $-\\frac{1}{3}, y+2, 4y, \\ldots$. Solve for $y$. \n,
```
************************************************************************
```
**Low RI**:  For an arithmetic sequence, the difference between consecutive terms must be equal. Therefore, we can set up the following equations based on the sequence given:\n\\[ (y + 2) - \\left(-\\frac{1}{3}\\right)

**Low KN**: For an arithmetic sequence, the difference between consecutive terms must be equal. Therefore, we can set up the following equations based on the sequence given:\n\\[ (y + 2) - \\left(-\\frac{1}{3}\\right)

**Low TR**:For an arithmetic sequence, the difference between consecutive terms must be equal. Therefore, we can set up the following equations based on the sequence given:\n\\[ (y + 2) - \\left(-\\frac{1}{3}\\right)

```
************************************************************************
```
**Input II**: Find all solutions to the equation $\\displaystyle\\sqrt[3]{3 - \\frac{x}{3}} = -2$. \n,
```
************************************************************************
```
**Low RI**: Start by isolating the cube root:\n$$ \\sqrt[3]{3 - \\frac{x}{3}} = -2 $$\n\nCube both sides to eliminate the cube root:\n$$ 3 - \\frac{x}{3} = (-2)^3 $$\n$$ 3 - \\frac{x}{3} = -8 $$\n\nSolve for $x$:\n$$ 3 + 8 = \\frac{x}{3} $$\n$$ 11 = \\frac{x}{3} $$\n$$ x = 33 $$\n\nThus, the solution to the equation is:\n$$ \\boxed{x = 33} $$

**Low KN**: Start by isolating the cube root:\n$$ \\sqrt[3]{3 - \\frac{x}{3}} = -2 $$\n\nCube both sides to eliminate the cube root:\n$$ 3 - \\frac{x}{3} = (-2)^3 $$\n$$ 3 - \\frac{x}{3} = -8 $$\n\nSolve for $x$:\n$$ 3 + 8 = \\frac{x}{3} $$\n$$ 11 = \\frac{x}{3} $$\n$$ x = 33 $$\n\nThus, the solution to the equation is:\n$$ \\boxed{x = 33} $$

**Low TR**: Start by isolating the cube root:\n$$ \\sqrt[3]{3 - \\frac{x}{3}} = -2 $$\n\nCube both sides to eliminate the cube root:\n$$ 3 - \\frac{x}{3} = (-2)^3 $$\n$$ 3 - \\frac{x}{3} = -8 $$\n\nSolve for $x$:\n$$ 3 + 8 = \\frac{x}{3} $$\n$$ 11 = \\frac{x}{3} $$\n$$ x = 33 $$\n\nThus, the solution to the equation is:\n$$ \\boxed{x = 33} $$

Figure 9: Low score tokens in NuminaMath-CoT dataset.

```
**********************************************************************
Input : Write a Python program to count the number of vowels in a given
string.
**********************************************************************
Low RI: ```python
def count_vowels(string):
    """
    This function takes a string as input and counts the number of vowels
in it.
    Parameters:
    string (str): The input string
    Returns:
    int: The number of vowels in the string
    """
    vowels = "aeiouAEIOU"  # Define a string of all vowels (both lowercase
and uppercase)
    count = 0  # Initialize a counter variable
    for char in string:  # Iterate over each character in the string
        if char in vowels:  # Check if the character is a vowel
            count += 1  # Increment the counter by 1
    return count  # Return the final count of vowels
Low KN: ```python
def count_vowels(string):
    """
    This function takes a string as input and counts the number of vowels
in it.
    Parameters:
    string (str): The input string
    Returns:
    int: The number of vowels in the string
    """
    vowels = "aeiouAEIOU"  # Define a string of all vowels (both lowercase
and uppercase)
    count = 0  # Initialize a counter variable
    for char in string:  # Iterate over each character in the string
        if char in vowels:  # Check if the character is a vowel
            count += 1  # Increment the counter by 1
    return count  # Return the final count of vowels
Low TR: ```python
def count_vowels(string):
    """
    This function takes a string as input and counts the number of vowels
in it.
    Parameters:
    string (str): The input string
    Returns:
    int: The number of vowels in the string
    """
    vowels = "aeiouAEIOU"  # Define a string of all vowels (both lowercase
and uppercase)
    count = 0  # Initialize a counter variable
    for char in string:  # Iterate over each character in the string
        if char in vowels:  # Check if the character is a vowel
            count += 1  # Increment the counter by 1

    return count  # Return the final count of vowels
```

Figure 10: Low score tokens in CodeExercise-Python-27k dataset.

\*\*\*\*\*\*\*\*\*\*\*\*\*\*\*\*\*\*\*\*\*\*\*\*\*\*\*\*\*\*\*\*\*\*\*\*\*\*\*\*\*\*\*\*\*\*\*\*\*\*\*\*\*\*\*\*\*\*\*\*\*\*\*\*\*\*\*\*\*\*\*\*\*\*\*\*\*\*\*\*\*

**Input** : Answer the question based on the following context: During augmentation and Mitrofanoff procedures, conduits are usually implanted into the posterior bladder wall. Anatomical considerations may necessitate an anterior conduit. To compare the relative drainage efficiency in patients with posterior and anterior conduits, we studied their rates of bladder stone formation and urinary tract infection (UTI). A retrospective chart review identified exstrophy patients who underwent augmentation and Mitrofanoff between 1991 and 2003. Patients with 3 years or greater follow-up were included. Fifty-four patients fit this criterion, with a conduit implanted anteriorly (33) or posteriorly (21). We compared rates of bladder stone formation and UTI. Stomal revisions and the status of the bladder neck were also noted. Stone formation and UTI rates were higher in the anterior conduits, although only UTI showed a statistically significant difference. Patient demographics were similar between the two groups, including age and sex. The rates of stomal complications and the bladder neck status were also similar.
Question: Mitrofanoff continent catheterizable conduits: top down or bottom up?

\*\*\*\*\*\*\*\*\*\*\*\*\*\*\*\*\*\*\*\*\*\*\*\*\*\*\*\*\*\*\*\*\*\*\*\*\*\*\*\*\*\*\*\*\*\*\*\*\*\*\*\*\*\*\*\*\*\*\*\*\*\*\*\*\*\*\*\*\*\*\*\*\*\*\*\*\*\*\*\*\*

**Low RI**: Patients with anterior conduits had an increased risk of UTI and bladder stone formation compared to those with posterior conduits, although this was not significant in the case of bladder stone rate. This may indicate sub-optimal bladder drainage and should be addressed with careful preoperative counseling and close follow-up.

**Low KN**: Patients with anterior conduits had an increased risk of UTI and bladder stone formation compared to those with posterior conduits, although this was not significant in the case of bladder stone rate. This may indicate sub-optimal bladder drainage and should be addressed with careful preoperative counseling and close follow-up.

**Low TR**: Patients with anterior conduits had an increased risk of UTI and bladder stone formation compared to those with posterior conduits, although this was not significant in the case of bladder stone rate. This may indicate sub-optimal bladder drainage and should be addressed with careful preoperative counseling and close follow-up.

Figure 11: Low score tokens in PubMedQA dataset.

\*\*\*\*\*\*\*\*\*\*\*\*\*\*\*\*\*\*\*\*\*\*\*\*\*\*\*\*\*\*\*\*\*\*\*\*\*\*\*\*\*\*\*\*\*\*\*\*\*\*\*\*\*\*\*\*\*\*\*\*\*\*\*\*\*\*\*\*\*\*\*\*\*\*\*\*\*\*\*\*\*\*\*

**Input** : Based on your financial expertise, provide your response or viewpoint on the given financial question or topic.\nDo I need to keep paper records for my business?,

output: Scanned or electronic copies of invoices should be sufficient as long as they are accurate and you can deliver them during an audit. Also, if you have an accountant prepare your taxes you would either need to provide them a copy of the invoices or a summary of them with the corresponding amounts to be claimed. Personally I prefer to print out a paper copy and file that away with that quarter's and year's other tax documents. I do my own taxes and find paper copies handy as I can go through each invoice/receipt and make sure I have entered its information by ticking it. I find that when handling a large number of documents that paper copies are more easy to handle than electronic ones. In the end you will need to use a system that you feel comfortable with and are able to use effectively.

\*\*\*\*\*\*\*\*\*\*\*\*\*\*\*\*\*\*\*\*\*\*\*\*\*\*\*\*\*\*\*\*\*\*\*\*\*\*\*\*\*\*\*\*\*\*\*\*\*\*\*\*\*\*\*\*\*\*\*\*\*\*\*\*\*\*\*\*\*\*\*\*\*\*\*\*\*\*\*\*\*\*\*

**Low RI**: Scanned or electronic copies of invoices should be sufficient as long as they are accurate and you can deliver them during an audit. Also, if you have an accountant prepare your taxes you would either need to provide them a copy of the invoices or a summary of them with the corresponding amounts to be claimed. Personally I prefer to print out a paper copy and file that away with that quarter's and year's other tax documents. I do my own taxes and find paper copies handy as I can go through each invoice/receipt and make sure I have entered its information by ticking it. I find that when handling a large number of documents that paper copies are more easy to handle than electronic ones. In the end you will need to use a system that you feel comfortable with and are able to use effectively.

**Low KN**: Scanned or electronic copies of invoices should be sufficient as long as they are accurate and you can deliver them during an audit. Also, if you have an accountant prepare your taxes you would either need to provide them a copy of the invoices or a summary of them with the corresponding amounts to be claimed. Personally I prefer to print out a paper copy and file that away with that quarter's and year's other tax documents. I do my own taxes and find paper copies handy as I can go through each invoice/receipt and make sure I have entered its information by ticking it. I find that when handling a large number of documents that paper copies are more easy to handle than electronic ones. In the end you will need to use a system that you feel comfortable with and are able to use effectively.

**Low TR**: Scanned or electronic copies of invoices should be sufficient as long as they are accurate and you can deliver them during an audit. Also, if you have an accountant prepare your taxes you would either need to provide them a copy of the invoices or a summary of them with the corresponding amounts to be claimed. Personally I prefer to print out a paper copy and file that away with that quarter's and year's other tax documents. I do my own taxes and find paper copies handy as I can go through each invoice/receipt and make sure I have entered its information by ticking it. I find that when handling a large number of documents that paper copies are more easy to handle than electronic ones. In the end you will need to use a system that you feel comfortable with and are able to use effectively.

Figure 12: Low score tokens in fiqa dataset.