# OpenReview forum: "Explainable Token-level Noise Filtering for LLM Fine-tuning Datasets"
_ICLR.cc/2026/Conference — ICLR 2026 Poster_

### Official Review · Reviewer_9uW5 · 2025-10-26

**Soundness:** 2
**Presentation:** 3
**Contribution:** 2
**Rating:** 4
**Confidence:** 4

**Summary:**

This paper addresses the mismatch between sentence-level labeling and token-level optimization in LLM fine-tuning. Most fine-tuning datasets provide labels at the sentence or document level, causing models to learn all tokens with equal importance. As a result, noisy or irrelevant tokens within a sentence can distort the optimization direction.
To resolve this issue, the authors propose XTF (eXplainable Token-level Filtering). XTF decomposes each token’s contribution into three interpretable attributes: Reasoning Importance (RI), Knowledge Novelty (KN), and Task Relevance (TR). Based on statistically derived thresholds, noisy tokens are identified and masked during training, ensuring that the model learns from meaningful tokens only.

Experiments across three downstream tasks and multiple LLM architectures demonstrate that XTF consistently outperforms both standard fine-tuning and existing data filtering methods. The authors also provide theoretical evidence showing that filtering improves gradient alignment and accelerates convergence. While token-level computation introduces additional overhead, the paper discusses potential efficiency improvements through model distillation.

Overall, this study presents a theoretically grounded and empirically validated framework for token-level data refinement, offering a practical and interpretable solution to one of the fundamental limitations in LLM fine-tuning.

**Strengths:**

S1. The three metrics are designed in a complementary manner, which is a notable strength of the paper. Knowledge Novelty (KN) increases as the model’s probability of correctly predicting the next token (PCP) decreases, but on its own, a low PCP could simply indicate meaningless noise. The authors address this by jointly considering Reasoning Importance (RI) and Task Relevance (TR), filtering out tokens with low RI or TR even if KN is high. Only when all three metrics are high is a token regarded as meaningful new knowledge. This complementary design effectively overcomes the limitations of each individual metric and demonstrates a distinctive approach to fine-grained token-level filtering.

S2. The paper takes a logical and rigorous approach by providing mathematical proofs that explain why the proposed filtering mechanism is theoretically effective. This theoretical grounding enhances the credibility of the method and clearly supports the intuition behind its practical usefulness.

**Weaknesses:**

W1. The core techniques are borrowed from existing methods without significant innovation: attention scores for importance, which is a standard interpretability technique, prediction probability, which is a straightforward application, and semantic distance, which is basic embedding similarity. The contribution is primarily engineering existing components rather than introducing novel algorithms or architectures.

W2. While motivated by score distributions (Fig 3), the specific thresholding methods involve some heuristics (e.g., the exact quantile parameter for RI, the fixed 95% PCP threshold for KN, choice of cluster for TR via Multi-Otsu). Sensitivity analysis (Appendix C.1) shows the chosen thresholds work best, but optimality across all scenarios isn't guaranteed.

W3. Ablations are missing. The paper needs: 1) threshold sensitivity, where Table 4 tests ±3% but not other ranges or methods, 2) attribute weighting, where Instead of union, can we weight attributes?, 3) layer selection, where does using different attention layers change results?, and 4) dataset size, where how does XTF perform with more/less training data?

W4. Calculating the three scores for all tokens in a dataset requires inference passes with the base LLM, adding a non-trivial computational cost before fine-tuning can begin. The paper quantifies this in Appendix C.2, but it could be a barrier for very large datasets or models.

**Questions:**

Q1. The paper uses distinct methods (Quantile, fixed threshold, Multi-Otsu) for thresholding the three scores based on observed distributions. Could a more unified approach, perhaps learning optimal thresholds or using a combined score, be feasible or beneficial?

Q2. Appendix C.2 shows scoring time. How does the total time (scoring + XTF fine-tuning) compare to achieving similar performance with baseline methods (e.g., normal fine-tuning for more epochs, or sample-level filtering + normal fine-tuning)?

Q3. Could you provide further justification or analysis on how the adaptive constant (r₁) for RI and the heuristic threshold (e.g., PCP ≥ 0.95) for KN were determined? In particular, it would be helpful if you could include additional discussion or experimental evidence showing how variations in these parameters influence model performance.

Q4. In the current study, the filtering ratio is adjusted in ±3% intervals to examine the effect of threshold changes, but this interval may appear somewhat arbitrary. Could you provide additional experimental results using finer intervals (e.g., ±1%) or a wider range of values to analyze the performance patterns and identify the potential optimal threshold range more precisely?

---

> ### Author Response · Authors · 2025-11-24
> **Response to 9uW5 -- Part I**
>
> **We sincerely appreciate your positive recognition of the framework design and the novelty of our paper, which has greatly encouraged us!** We hereby provide responses to the questions you raise in this context, clarify any potential misunderstanding, and sincerely hope that you adjust the rating of this work accordingly.
>
> ---
>
> **Q1:** The paper uses distinct methods (Quantile, fixed threshold, Multi-Otsu) for thresholding the three scores based on observed distributions. Could a more unified approach, perhaps learning optimal thresholds or using a combined score, be feasible or beneficial?
>
> **Re:** Thank you for pointing out the potential directions for optimizing XTF. We consider this issue carefully and regard it as infeasible. The complementarity among multi-dimensional attributes is a major advantage of XTF, and it allows us to use conservative threshold design to protect tokens that are not highly valuable but still have a positive impact on training. Using an integrated score, for example by summing the three scores, **introduces complex transferability challenges for threshold selection when switching models and datasets.** At the same time, as discussed in Section 3.1, we avoid integrated scoring because its justification cannot be supported: this arises from the fundamental difficulty created by computational cost and the scale gap between the training dataset and individual tokens, and it is also the challenge that the XTF framework attempts to simplify. What's more, learning or optimizing the best thresholds is unlikely to generalize across different models and training tasks, and targeted training would **significantly increase computational cost**, which is not an appropriate choice.
>
> ___
>
> **Q2:** Appendix C.2 shows scoring time. How does the total time (scoring + XTF fine-tuning) compare to achieving similar performance with baseline methods (e.g., normal fine-tuning for more epochs, or sample-level filtering + normal fine-tuning)?
>
> **Re:** As stated in Appendix D, **the consumption varies significantly with changes in the data scale and sample length**. Providing a direct time comparison may not be sufficiently convincing. However, compared with recent methods designed for the same task, computational cost is an advantage of XTF. The methods such as SLM (or Rho) and Token-clean based on reference models, incur computational costs that far exceed those of XTF. These methods require model training, whereas XTF requires only inference-level computational cost, which is similar to sample-level methods such as DA and DF. We also appreciate your reminder, and we **include a more detailed discussion of computational cost comparation in Appendix D, marked in blue.**
>
> ---
>
> **Q3:** Could you provide further justification or analysis on how the adaptive constant (r₁) for RI and the heuristic threshold (e.g., PCP ≥ 0.95) for KN were determined? In particular, it would be helpful if you could include additional discussion or experimental evidence showing how variations in these parameters influence model performance.
>
> **Re:** Thank you for pointing out the issues regarding parameter settings in the scoring system. In our experiments, the IQR constant is set to 1 and we also recommend setting it to 1, as this avoids excessive token filtering when attention exhibits a binary distribution. **To prevent potential misunderstandings, we modify the original equation (5) by removing the adaptive parameter r1.** The 95% threshold for KN is determined empirically. In the new version, **we add a discussion of threshold analysis (Section 3.3 and Appendix C.2).** We aim to provide a flexible and general threshold-selection criterion, rather than presenting fixed-percentage results such as 5% or 10% as in existing baselines (SLM/RHO, DF). The flexibility of the XTF framework supports this approach: **it requires only the conservativeness of threshold selection (it must not be too strict), unlike single-metric methods, which require precision in threshold selection (it must be neither too high nor too low)**. Our latest threshold analysis in Section 3.3 explains the advantages enabled by this multi-attribute complementarity.

---

> > ### Author Response · Authors · 2025-11-24
> > **Response to 9uW5 -- Part II**
> >
> > ---
> >
> > **Q4:**  In the current study, the filtering ratio is adjusted in ±3% intervals to examine the effect of threshold changes, but this interval may appear somewhat arbitrary. Could you provide additional experimental results using finer intervals (e.g., ±1%) or a wider range of values to analyze the performance patterns and identify the potential optimal threshold range more precisely?
> >
> > **Re:** Thank you for raising concerns regarding threshold perturbation. We are pleased to provide this explanation, which is missing in the paper. A perturbation range of 3% is appropriate for our setting because the number of filtered tokens in a sentence is relatively small. If a perturbation of 1–2% is used, the filtered tokens in many sentences **remain completely unchanged**. A value of 3% is therefore a reasonable choice based on the experimental setting and granularity. **We clarify this point in the new version of Appendix C.2.**
> >
> > ---
> >
> > **Once again, thank you for your time and effort! All newly added content in the paper will be highlighted in blue and uploaded. We hope our responses can clarify any potential misunderstandings and address your concerns.**

---

> ### Author Response · Authors · 2025-11-28
> **Thanks to Reviewer 9uW5**
>
> Please allow us to once again express our sincere gratitude to you. The time and effort you have dedicated to the review process are of immense value to our work. Your recognition of the theoretical analysis and framework designing in our research is truly encouraging. The suggestions you provided, such as improving the baseline, adding threshold analysis and overhead analysis, have significantly improved the quality of our work.
>
> Please let us know if our response and the new experiments have properly addressed your concerns. We are more than happy to answer any additional questions during the rebuttal period. Your feedback will be greatly appreciated.

---

### Official Review · Reviewer_XoFh · 2025-10-29

**Soundness:** 3
**Presentation:** 2
**Contribution:** 2
**Rating:** 4
**Confidence:** 4

**Summary:**

This paper investigates the token-level noise problem in fine-tuning datasets, motivated by recent findings that noisy tokens can significantly degrade model performance. To address this issue, the paper proposes XTF, a token-level noise filtering framework that identifies and removes noisy tokens based on three key attributes: reasoning importance, knowledge novelty, and task relevance. These attributes are quantified using the attention score, PCP score, and relevance score, respectively. Extensive experiments across multiple base models and benchmarks demonstrate that XTF consistently outperforms all baseline methods.

**Strengths:**

- The writing is clear and easy to follow.

- The paper lies in introducing a noisy token filtering framework that jointly leverages several scores (including attention, PCP, relevance scores) to accurately identify and remove noisy tokens, which is novel and interesting.

- The claims are well-supported by empirical experiments across multiple models (LLaMA-3, Mistral, Deepseek-distilled-qwen), and popular evaluation benchmarks.

**Weaknesses:**

-  The authors fail to provide a detailed comparison of computational overhead with existing baselines, such as SLM. Including such analysis would help clarify the efficiency of the proposed method.

-  The benchmarks used in the paper are relatively weak and do not convincingly demonstrate the method’s effectiveness. For instance, in Figure 4, the baselines—namely, the original base model (CA) and the standard fine-tuning version (Normal)—are too naive. The reviewer suggests incorporating more competitive baselines for comparison. Recent work such as [1] provides a strong theoretical foundation for noisy token cleaning and should be considered as an additional baseline.

- The paper would benefit from evaluation on more diverse and challenging datasets to further validate its effectiveness—for example, BBH, ARC, and MMLU, which are widely recognized benchmarks for reasoning and knowledge assessment.


[1] Token Cleaning: Fine-Grained Data Selection for LLM Supervised Fine-Tuning, ICML 2025.

**Questions:**

1. The citation format throughout the paper is inconsistent with standard academic conventions. Please revise all references to follow the correct style

2. A reference is missing in Line 101.

3. Could incorporating additional attributes further improve the accuracy of noisy token identification?

4. The effectiveness of SLM largely depends on the quality of its reference model. For SLM (also known as RHO), how many samples were used to train the reference model?

5. Could you provide a detailed comparison of GPU hours between the proposed method and the baselines to clearly illustrate the computational overhead?

---

> ### Author Response · Authors · 2025-11-24
> **Response to Reviewer XoFh**
>
> **We sincerely appreciate your positive recognition of the writing, the novelty, and the experiment of our paper, which has greatly encouraged us!** We hereby provide responses to the questions you raise in this context, clarify any potential misunderstanding, and sincerely hope that you adjust the rating of this work accordingly.
>
> ---
>
> **Q1:** The citation format throughout the paper is inconsistent with standard academic conventions. Please revise all references to follow the correct style.
>
> **Q2:** A reference is missing in Line 101.
>
> **Re:** Thank you for your suggestions regarding the paper format. Our citation style follows the standard ICLR26 template, but some citation entries may not sufficiently consistent. Following your reminder, we standardize the citation entries and correct the typos.
>
> ---
>
> **Q3:** Could incorporating additional attributes further improve the accuracy of noisy token identification?
>
> **Re:** You are correct that incorporating additional potential attributes is likely to further improve the final performance of XTF. XTF adopts an aggressive filtering strategy (taking the union) and a conservative threshold-selection strategy (a relatively low filtering threshold), **which enables compatibility across different attributes and ensures the rationality of the filtering process**. In future work, identifying more potential attributes or performing targeted optimization of the scoring system will be supported by the XTF framework and represents a promising direction to explore.
>
> ---
>
> **Q4:** The effectiveness of SLM largely depends on the quality of its reference model. For SLM (also known as RHO), how many samples were used to train the reference model?
>
> **Re:** Thank you for your detailed question. Originally, the SLM used "high-quality data" when training the reference model, which is not present in the real-world SFT setting. To address this, we refer to the work Token-Clean, which employs a method very similar to SLM and uses the whole original training dataset to obtain the reference model. In the new version, we also include Token-Clean (using its self-evolve version of the open-source code) as a baseline to enrich our experiments.
>
> ---
>
> **Q5:** Could you provide a detailed comparison of GPU hours between the proposed method and the baselines to clearly illustrate the computational overhead?
>
> **Re:** As stated in Appendix D, the consumption varies significantly with changes in the data volume and content. Providing a direct time comparison may not be sufficiently convincing. However, **compared with recent methods designed for the same task, computational cost is an advantage of XTF**. The methods such as SLM (or Rho) and Token-clean based on reference models, incur computational costs that far exceed those of XTF. **These methods require model training, whereas XTF requires only inference-level computational cost**, which is similar to sample-level methods such as DA and DF. We also appreciate your reminder, and we include a more detailed discussion of computational cost comparation in Appendix D, marked in blue.
>
> ---
>
> In addition to the modifications described above, we also **improve the sufficiency of the experiments (the issue you mentioned in the weakness section)**. Since datasets such as BBH, ARC, and MMLU are mostly in multiple-choice format, i.e., they do not require optimization with respect to the output label, they are not fully aligned with the scenario targeted by our method. Therefore, we expand the experiments using datasets whose output labels are sentences (NuminaMath-CoT, MATH-500, and FIQA). **The additional experiments are shown in Appendix C.1.**
>
> ---
>
> **Once again, thank you for your time and effort!  All newly added content in the paper will be highlighted in blue and uploaded. We hope our responses can clarify any potential misunderstandings and address your concerns.**

---

> ### Author Response · Authors · 2025-11-28
> **Thanks to Reviewer XoFh**
>
> Please allow us to once again express our sincere gratitude to you. The time and effort you have dedicated to the review process are of immense value to our work. Your recognition of the paper writing, framework designing and the experiments in our research is truly encouraging. The suggestions you provided, such as expanding the experiments and adding overhead analysis, have significantly improved the quality of our work.
>
> Please let us know if our response and the new experiments have properly addressed your concerns. We are more than happy to answer any additional questions during the rebuttal period. Your feedback will be greatly appreciated.

---

### Official Review · Reviewer_Ucn3 · 2025-10-29

**Soundness:** 3
**Presentation:** 3
**Contribution:** 2
**Rating:** 6
**Confidence:** 4

**Summary:**

This paper introduces XTF, a novel learning framework that enhances Large Language Model (LLM) fine-tuning by filtering token-level training data. XTF identifies and removes noisy tokens based on three proposed metrics: reasoning importance, knowledge novelty, and task relevance. Tokens scoring below a threshold on any metric are considered noise, and their loss contribution is blocked during backpropagation. Extensive experiments on mathematical, coding, and medical tasks across seven different LLMs demonstrate the strong performance of the proposed framework.

**Strengths:**

-  Unlike traditional token-level filtering methods that rely solely on loss values, Authors propose three distinct metrics: reasoning importance, knowledge novelty, and task relevance, to comprehensively assess each token's contribution. These metrics are designed to capture different aspects of the relationship between the model, the training data, and the target task.

- Theoretical analysis are provided to demonstrates the effectiveness and efficiency of the XTF framework.

- Extensive experiments across three downstream domains (math, code, and medicine) and seven LLMs confirm the strong performance of our approach.

**Weaknesses:**

- The threshold selection process for the three metrics appears to be largely empirical. For instance, the use of a quantile method in Equation (5) and a fixed threshold of 0.05 in Equation (6). The generalizability of these specific threshold needs further investigation.

- The paper posits that a lower PCP indicates a token containing novel knowledge. However, a low score can also reflect an incorrect or nonsensical token. This ambiguity creates a paradox that is not addressed: how can the framework reliably distinguish between "novel knowledge" and "noise"?

- The empirical evaluation, while demonstrating positive results, is conducted on a limited number of datasets (three). This contrasts with related work in the field (e.g., [R1, R2]), which often employs more extensive benchmarks (e.g., seven datasets) to validate generalizability.

- Typos. "Section ??" on Line 101, should be corrected.

[R1] RHO-1: Not All Tokens Are What You Need

[R2] Token Cleaning: Fine-Grained Data Selection for LLM Supervised Fine-Tuning

**Questions:**

1. The threshold selection process requires deeper analysis. A sensitivity analysis exploring different threshold values is needed to justify the current choices.

2. To firmly establish XTF's superiority, evaluations should be expanded to include more datasets and a broader comparison with baselines.

---

> ### Author Response · Authors · 2025-11-24
> **Response to Reviewer Ucn3**
>
> **We sincerely appreciate your positive recognition of the novelty, the experimental results, and the theoretical analysis of our work, which has greatly encouraged us!** We hereby provide responses to the questions you raise in this context, clarify any potential misunderstanding, and sincerely hope that you adjust the rating of this work accordingly.
>
> ---
>
> **Q1:** The threshold selection process requires deeper analysis. A sensitivity analysis exploring different threshold values is needed to justify the current choices.
>
> **Re:**  Thank you for your suggestion. We have added more experimental results on threshold selection in the new paper version, refined the original analysis, included **perturbation experiments on numerical threshold variation (in Appendix C.2)**, and presented **the complementarity among tokens filtered by the three dimensions** (in Section 3.3).
>
> ---
>
> **Q2:** To firmly establish XTF's superiority, evaluations should be expanded to include more datasets and a broader comparison with baselines.
>
> **Re:**  Thank you for your suggestion. In order to demonstrate the generality of our method, we prioritize selecting a diverse range of dataset types (mathematics, medicine, and code) and a wide variety of training settings (different model scales and families, as well as both LoRA and full-parameter finetuning). As a result, we initially overlooked the number of datasets used, and we sincerely apologize for this omission. We now include additional experiments to strengthen the completeness of our evaluation. **In Appendix C.1, we add further results addressing experimental sufficiency**, including experiments on three additional datasets (NuminaMath-CoT, MATH-500, and FIQA). **We also incorporate comparisons with the token-clean baseline across all experiments**, which further demonstrates the generality of our method.
>
> ---
>
> **In addition to the modifications described above, we would also like to respond to the weakness you raised regarding threshold selection.**  As described in Section 3.1, **the quality of finetuning outcomes cannot be attributed at a fine-grained level to individual tokens**; that is, we cannot obtain fine-grained feedback on token filtering, which is the primary obstacle to threshold selection based on optimization methods. However, **within the XTF framework, the impact of this limitation is minimized**. Our strategy filters only the tokens that “completely lack’’ a given attribute, which allows us to use “conservative’’ thresholds to avoid mistakenly filtering valuable tokens and to maintain both the quantity and the quality of filtering through the compatible integration of multiple dimensions. Therefore, threshold selection may be empirical and imprecise; as long as it does not filter out valuable tokens, it remains acceptable. This is fundamentally different from single-metric methods (for example, rho [R1] and token clean [R2], which rely on changes in token loss). **When filtering is based on a single metric, threshold selection must be cautious and requires extensive experimentation to ensure generality**, whereas within the XTF framework, threshold selection only needs to ensure its conservativeness. On this basis, the XTF framework has the potential to support the discovery of more potential filtering dimensions in the future and their stable, non-interfering integration with the existing three dimensions, enabling strong extensibility.
>
> **At the same time, we also appreciate your sharp insight regarding the “PCP’’ metric.** In fact, during the early stages of developing the XTF framework, we did attempt to filter tokens using both “extremely high’’ and “extremely low’’ PCP thresholds. However, we find that filtering tokens with “extremely low’’ PCP conflicts with the “conservative’’ strategy of XTF, because **these tokens include not only illogical tokens but also knowledge that is “completely unknown’’ to the model**, and these two cases cannot be distinguished by the PCP score. In contrast, tokens with “extremely high’’ PCP consistently exhibit low training value and are often repetitive tokens or symbolic templates (as shown in Appendix E). **Fortunately, this does not imply that illogical tokens with extremely low PCP cannot be distinguished from valuable tokens**: RI scores can filter meaningless gibberish, and TR scores can identify incorrect word usage. This complementary effect across different dimensions is a strength of the XTF framework.
>
> ---
>
> **Once again, thank you for your time and effort! Your positive rating is truly encouraging to us. We hope our responses can clarify any potential misunderstandings and address your concerns.**

---

> ### Author Response · Authors · 2025-11-28
> **Thanks to Reviewer Ucn3**
>
> Please allow us to once again express our sincere gratitude to you. The time and effort you have dedicated to the review process are of immense value to our work. Your recognition of the novelty, theoretical analysis, and the experiments in our research is truly encouraging. The suggestions you provided, such as expanding the experiments and adding threshold analysis, have significantly improved the quality of our work.
>
> Please let us know if our response and the new experiments have properly addressed your concerns. We are more than happy to answer any additional questions during the rebuttal period. Your feedback will be greatly appreciated.

---

### Official Review · Reviewer_GsNo · 2025-10-31

**Soundness:** 2
**Presentation:** 3
**Contribution:** 2
**Rating:** 4
**Confidence:** 3

**Summary:**

This paper proposes a method called XTF (Explainable Token-level Noise Filtering) to improve fine-tuning for large language models (LLMs). The authors argue that most fine-tuning datasets are designed at the sentence level, while LLMs are optimized at the token level. This mismatch introduces token-level noise that can hurt model performance. The paper brings out three interpretable attributes: reasoning importance, knowledge novelty, and task relevance. Before fine-tuning the model, it will mask the one with the lowest score. This method is tested on three downstream tasks and it shows gaining accuracy compared to normal fine-tuning.

**Strengths:**

**Novel perspective:** The paper focuses on token-level dataset optimization, a topic that has not been well studied in LLM fine-tuning.

**Explainable approach:** The design and introduction of three attributes provides a clear and interpretable way to analyze which tokens are useful.

**Good experiment results:** The authors test on diverse tasks (math, medicine, code) and multiple model sizes, showing stable improvements.

**Good theoretical support:** Appendix A provides formal justification linking token filtering to gradient alignment, which adds credibility.

**Weaknesses:**

**Could do more on stable analysis:** Although results show improvements on accuracy, we don’t know whether the result is chosen based on the best result the experiment gets and whether it is reproducible. Thus, it could be better to include some variance, confidence intervals, or significance tests. It is hard to see how stable the improvements are.

**Could do more on examples:** Although the paper repeatedly emphasizes “explainable” token-level filtering, there is almost no qualitative examples or visualizations showing which tokens were removed.

**Questions:**

Threshold analysis is good but could do more in the future: It is good to see that the authors already include the threshold ablation in Appendix. A little bit of suggestion on this will be analyzing interactions effects between the three thresholds to see whether there is a trade-off or something, also it will be better to make some more discussion on this visible in the main paper.

How sensitive is the final performance when calculating three attributes in a specific way? For example, how would the different interaction among them affect the result?

Could the paper provide more specific examples on the process of filtering tokens?

How stable or consistent are these improvements the paper mention?

---

> ### Author Response · Authors · 2025-11-24
> **Response to Reviewer GsNo**
>
> **We sincerely appreciate your positive recognition of the novelty of our work, the theoretical analysis, and the experimental results, which has greatly encouraged us!** We hereby provide responses to the questions you raise in this context, clarify any potential misunderstanding, and sincerely hope that you adjust the rating of this work accordingly.
>
> ---
>
> **Q1:** Threshold analysis is good but could do more in the future: It is good to see that the authors already include the threshold ablation in Appendix. A little bit of suggestion on this will be analyzing interactions effects between the three thresholds to see whether there is a trade-off or something, also it will be better to make some more discussion on this visible in the main paper.
>
> **Re:** Thank you for your suggestion regarding threshold analysis, and your positive evaluation is highly encouraging to us. **We value your suggestion and add additional threshold-analysis content to the paper**, including **numerical perturbation analysis of thresholds (in Appendix C.2)** and **complementarity analysis across thresholds (in Section 3.3)**, in order to demonstrate the underlying logic of threshold selection in the XTF framework: a conservative threshold that accommodates multi-attribute compatibility.
>
> ---
>
> **Q2:** How sensitive is the final performance when calculating three attributes in a specific way? For example, how would the different interaction among them affect the result?
>
> **Re:** The three attributes are **complementary**, and the use of conservative thresholds ensures that the final results obtained after integrating the three attributes remain positive and beneficial. In the new content in Section 3.3, we provide a discussion of the complementarity in threshold analysis.
>
> ---
>
> **Q3:** Can the paper provide more specific examples regarding the filtering token process?
>
> **Re:** Thank you for your suggestion regarding the examples. In the new version, we have added a link to examples (Appendix E) in Section 3.3 of the main text. Our examples are originally provided in Appendix E, including two mathematics examples, one code example, and one medicine example, and each example contains the input data and the filtering results for all three dimensions. Following your suggestion, we include additional examples.
>
> ---
>
> **Q4:** How stable or consistent are these improvements mentioned in the paper?
>
> **Re:** Thank you for your suggestion regarding stability analysis. Since finetuning large models requires substantial computational resources (particularly for the commonly used and relatively large 7B, 8B, and 14B models), we are unable to conduct a sufficient number of repeated experiments for every training scenario to compute variances or confidence intervals, and we sincerely apologize for this limitation. However, **we adopt rigorous parameter configurations to ensure experimental fairness**. Specifically, we sample exactly the same training, validation, and test sets for all experiments and use fully transparent hyperparameters (shown in Table 3). In our practice, when models are trained with identical training data and training parameters, the final results exhibit only minimal deviation. Under this setup, the fairness of the training outcomes is well supported, and the results across different training scenarios (10 types) consistently demonstrate the general advantage of our method. We affirm that no selective reporting of results occurs.
>
> ---
>
> **Once again, thank you for your time and effort! All newly added content in the paper will be highlighted in blue and uploaded. We hope our responses can clarify any potential misunderstandings and address your concerns.**

---

> ### Author Response · Authors · 2025-11-28
> **Thanks to Reviewer GsNo**
>
> Please allow us to once again express our sincere gratitude to you. The time and effort you have dedicated to the review process are of immense value to our work. Your recognition of the novelty, theoretical analysis, and the experimental results in our research is truly encouraging. The suggestions you provided, such as adding explainable examples and threshold analysis, have significantly improved the quality of our work.
>
> Please let us know if our response and the new experiments have properly addressed your concerns. We are more than happy to answer any additional questions during the rebuttal period. Your feedback will be greatly appreciated.

---

### Author Response · Authors · 2025-12-02
**Rebuttal Summary**

**We sincerely appreciate the valuable time and effort of the reviewers, AC, SAC, and PC. All of your feedback has significantly contributed to improving the quality of our work!**

 As the rebuttal process approaches its conclusion, we would like to provide a concise summary, particularly **highlighting the newly added content**, to assist the AC in making the final decision.

The improvements to the paper can be categorized into **four** main parts:

---

**1. Threshold Analysis**:

In response to concerns **raised by reviewers GsNo, Ucn3, and 9uW5** regarding the threshold, we have added corresponding experiments. Specifically, we have introduced **Threshold Analysis part in Section 3.3**, added **Filtered Tokens in Appendix C.2**, and added **Figures 4 and 6**. Together with the previously existing content (Figures 3, Table 5, and Figure 7), we believe the updated threshold analysis can addresses the reviewers' concerns.

---

**2. Experimental Sufficiency and Baseline Enhancement**

In response to **feedback from reviewers Ucn3 and XoFh** regarding insufficient experimental datasets and the absence of SOTA baseline comparisons, we have expanded the main experiments. Specifically, we have **added Appendix C.1**, which extends the main experiment using three datasets to construct two sets of experiments (shown in the **newly added Table 4**). Additionally, we have incorporated the **Token-cleaning baseline**, as recommended by the reviewers, into all the main experiments. We believe these additions can address concerns about the sufficiency of the experiments.

---

**3. Computational Cost Analysis**

In response to comments **from reviewers XoFh and 9uW5** regarding computational cost comparisons, we have included  **a new part Cost Analysis in Appendix D**. We have chosen not to provide direct data comparisons, as this could be misleading in industrial settings, where different fine-tuning strategies or data processing methods are employed. Instead, we have **analyzed the computational costs from a perspective of computational complexity** and supported our arguments by the significant gaps in training and inference costs.

---

**4. Specific Examples**

**Reviewer GsNo mentioned** the lack of concrete examples and visualizations (*he/she may have been overlooked the examples provided in original manuscript*, due to issues in the organization of our paper). In the revised version, we have added **more specific examples (Appendix E)** and provided **explicit \ref link in Section 3.3** of the main text.

---

In addition to these four key areas, we have also response other concerns raised by the reviewers carefully and made corresponding revisions to the paper. **For issues mentioned in the weaknesses but not in the questions**, we have provided additional clarifications in our responses.

---

We have dedicated considerable effort to this paper, where we **conducted additional experiments and made numerous revisions**. Given the special policies implemented this year to address unforeseen circumstances, **we sincerely hope that the Area Chair will take our rebuttal and the new revisions into account when making the final decision**.

**Once again, we express our heartfelt gratitude for your hard work and support!**

---

### Meta-Review · Area_Chair_pfWp · 2026-01-06

**Summary:**

The paper provides a technique for filtering tokens for LLM fine-tuning. The authors provide an approach based on an analysis of what could cause a token to hurt the fine-tuning process. They define 3 separate issues that might cause a token to be hurtful / noisy, and devise a filtering strategy based on 3 separate criteria. The reviews showed a positive attitude towards the novelty and sensibility of this approach. This seems like the main strength of the paper: GsNo, “The design and introduction of three attributes provides a clear and interpretable way to analyze which tokens are useful”, Ucn3 “Unlike traditional token-level filtering methods that rely solely on loss values, Authors propose three distinct metrics…”, 9uW5 “The three metrics are designed in a complementary manner, which is a notable strength of the paper.”

In addition to the sensibility of the approach, the reviews appreciated the theoretical analysis backing the used techniques, and the thorough experiments demonstrating the effectiveness of technique over multiple LLMs in different scenarios.
The last point was not mentioned purely as a strength, and some reviews ask for additional evaluations, either in terms of other data sets to be used for fine-tuning, and additional baselines. The authors provided experiments on two additional dataset (NuminaMath-CoT, FIQA), and added an additional baseline “token cleaning” (published in ICML ‘25). The new experiments follow the same trends as the original ones showing the advantage of the proposed method, and mitigating the concerns regarding additional bottom-line experiments.

Another notable concern raised by more than one review is the thresholding method. The method provided in the paper is quite heuristic and the reviews asked for either a justification for this thresholding method, or some analysis proving its generalizability. The authors provided new experiments where the modify the thresholds by +- 3%, and show that the selected heuristics provide the best setting out of these perturbed choices. The authors deduce from this that their thresholding methods are robust. Personally, I do acknowledge that these experiments show the thresholding does have some merit, but it also shows the lack of stability for this hyperparameter. These slight perturbations (the authors mentioned <3% was hard to implement since it is too small) cause a large change (roughly 5%) in downstream performance, leading me to conclude that the thresholding is a weakness of the paper.

The reviews raised a few other concerns but they were minor compared to those mentioned above. Considering the pros vs cons for the paper, we have a paper that is agreed to provide a novel, sensible and theory backed method, showing an improvement over SoTA techniques in a wide range of settings (originally a concern, likely the cause of some of the low scores, but mitigated in the rebuttal). The main limitation of it is the sensitivity of the thresholding. While it is not something to be ignored, to me it seems that the paper can still be a good addition despite this limitation, as long as it is presented in an honest way.

**Reviewer Concerns:**

The main concern was the request for an additional baseline and usecase, and it was fully mitigated in the rebuttal.
The second concern of sensitivity to the threshold (a hyperparameter) was only partially resolved, but this is not a concern blocking the paper from being accepted, rather its an acceptable limitation

**Reviewer Scores:**

Ucn3 and XoFh mentioned the need for additional baselines and usecases, and since this was provided it is quite possible they would have raised their score to 6 or 8.
The concerns raised by 9uW5 and GsNo where not fully resolved, but reading GsNo's review, it seems more like an accept than a reject, so given the partial resolution of the concerns and other reviews they might have also raised the score from 4 to 6.

---

### Decision · Program_Chairs · 2026-01-26

Accept (Poster)